# Advances in the Fabrication and Characterization of Superhydrophobic Surfaces Inspired by the Lotus Leaf

**DOI:** 10.3390/biomimetics7040196

**Published:** 2022-11-10

**Authors:** Melika Farzam, Mohamadreza Beitollahpoor, Samuel E. Solomon, Henry S. Ashbaugh, Noshir S. Pesika

**Affiliations:** Chemical and Biomolecular Engineering Department, Tulane University, New Orleans, LA 70118, USA

**Keywords:** biomimetics, contact angle, contact angle hysteresis, surface wettability, liquid friction, Young’s equation

## Abstract

Nature has proven to be a valuable resource in inspiring the development of novel technologies. The field of biomimetics emerged centuries ago as scientists sought to understand the fundamental science behind the extraordinary properties of organisms in nature and applied the new science to mimic a desired property using various materials. Through evolution, living organisms have developed specialized surface coatings and chemistries with extraordinary properties such as the superhydrophobicity, which has been exploited to maintain structural integrity and for survival in harsh environments. The Lotus leaf is one of many examples which has inspired the fabrication of superhydrophobic surfaces. In this review, the fundamental science, supported by rigorous derivations from a thermodynamic perspective, is presented to explain the origin of superhydrophobicity. Based on theory, the interplay between surface morphology and chemistry is shown to influence surface wetting properties of materials. Various fabrication techniques to create superhydrophobic surfaces are also presented along with the corresponding advantages and/or disadvantages. Recent advances in the characterization techniques used to quantify the superhydrophobicity of surfaces is presented with respect to accuracy and sensitivity of the measurements. Challenges associated with the fabrication and characterization of superhydrophobic surfaces are also discussed.

## 1. Introduction

The wetting of solid surfaces with various liquids is ubiquitous in our daily lives, from water drops sliding down a window on a rainy day to beading up on a rose petal or even completely bouncing off the wing of a butterfly. In nature, living organisms have evolved intricate surface chemistries and morphologies resulting in superior properties [1] required for survival. For example: some desert beetles use their structured fused wings to collect water from wind-driven morning fog [2]; the wings of butterflies and dragonflies are water-repellent, which allow them to survive in constant rainfall environments [3]; the water repellency of water strider (*Gerridae*) legs allows the insect to walk on water [4]; and emperor penguin (*Aptenodytes forsteri*) feathers trap air and provide insulation against the harsh weather in Antarctica [5].

In several of these examples, the surface relies on superhydrophobicity and/or water repellency (see Figure 1 [6,7,8]), i.e., surfaces with a static water contact angle (WCA), *θ*, of more than 150° and a sliding angle (SA) of less than 10° [9,10]. Superhydrophobic (SH) surfaces typically possess hierarchical micro- and nano-roughness and consist of low surface (or interfacial) energy materials or coatings. Inspired by the novel properties imbued by SH surfaces, researchers have proposed attractive applications of such surfaces in engineering technologies. Some of these biomimetic applications are in oil–water separation [11,12], corrosion and fouling resistance [13], anti-icing [14], drag reduction [15], self-cleaning [16,17], and antibacterial surfaces [18]. Over the years, there has been a growing interest in this field, which is evidenced by the increasing number of publications on this subject as depicted in Figure 2. Although the fabrication of SH surfaces can be traced back to 1907 in work by Ollivier [19], the concept of superhydrophobicity did not gain significant attention until this phenomenon was described in lotus leaves (*Nelumbo nucifera*) [20]. The fundamental science behind superhydrophobicity is generally well understood, however, researchers are still actively exploring novel fabrication techniques that are adaptable to metal, polymer, and ceramic surfaces and that can also increase the durability of SH surfaces or coatings against mechanical or chemical degradation [21,22,23]. Polymers have specifically played an important role; SH surfaces have either been fabricated directly from inherently hydrophobic polymers or polymers have been used to create SH coatings on various materials.

The main thrust of this review article is to provide a detailed understanding of the origin of superhydrophobicity, from the derivation of critical equations from a thermodynamic perspective to appreciating how surface chemistry and surface topography influence the performance of SH surfaces. Section 3 provides an overview of recent, and the most common, methods of fabricating SH surfaces along with their advantages and disadvantages. Section 4 provides an overview of the various techniques used to characterize SH surfaces. Based on these recent developments, we conclude with a summary of the main limitations that impede the development and use of SH surfaces in practical applications.

## 2. Surface Wetting and Water Repellency

A common method to quantify the wetting properties of surfaces is through contact angle measurements in which a small drop of a liquid is placed on a surface and the extent of wetting is quantified though the spreading of the drop. The Young’s equation [24] allows one to relate the observed contact angle that the liquid makes with the surface to interfacial energies within the system. A simple derivation of Young’s equation [24] can be obtained by considering a force balance at the 3-phase contact line, as shown in Figure 3.

By considering the interfacial energies of each pair of phases, i.e., $\gamma_{SL}$,$\;\gamma_{SV}$, and $\gamma_{LV}$ as forces acting over the wetted perimeter (or contact line) P of a water drop on a smooth solid surface (i.e., units of mN/m), a force balance yields Young’s equation,
(1)$$\gamma_{SV}=\gamma_{SL}+\gamma_{LV}\cos\theta ,$$
where  θ is the liquid contact angle, and $\gamma_{LV}\cos\theta$ is the component of $\gamma_{LV}$ resolved in the horizontal direction.

Young’s Equation (1) can also be derived by considering an energy balance in which a system consisting of a liquid drop of fixed volume in contact with a smooth solid surface surrounded by a vapor phase reaches equilibrium (i.e., the free energy *G* is a minimum) as shown in Figure 4. The free energy, *G*, of the system is a summation of the surface energies of the three interfaces,
(2)$$G=\gamma_{LV}A_{LV}+\gamma_{SL}A_{SL}+\gamma_{SV}A_{SV}$$
where $A_{LV}$, $\;A_{SL}$ and $A_{SV}$ are the surface areas of the liquid–vapor, solid–liquid, solid–vapor interfaces, respectively. Assuming a sufficiently small drop volume whereby gravitational distortions of the liquid–vapor interface can be neglected, $A_{LV}$ is equal to the area of a spherical cap, i.e.,
(3)$$A_{LV}=2\pi R^{2}\left(1-\cos\;\theta \right),$$
(4)$$A_{SL}=\pi X^{2}=\pi R^{2}\left(1-\cos^{2}\;\theta \right),$$
and
(5)$$A_{SV}=A_{S,total}-\pi R^{2}\left(1-\cos^{2}\;\theta \right)\;$$
where $A_{S,total}=A_{SV}+A_{SL}$ is the total solid surface area and remains constant within the system. Substituting Equations (3)–(5) into Equation (2) yields
(6)$$G=\gamma_{LV}2\pi R^{2}\left(1-\cos\;\theta \right)+(\gamma_{SL}-\gamma_{SV})\pi R^{2}\left(1-\cos^{2}\;\theta \right)+\gamma_{SV}A_{S,total}$$

To simplify the remainder of the derivation, we denote u=cosθ. Thus, rewriting Equation (6) yields
(7)$$G=\gamma_{LV}2\pi R^{2}\left(1-u\right)+(\gamma_{SL}-\gamma_{SV})\pi R^{2}\left(1-u^{2}\right)+\gamma_{SV}A_{S,total}$$

The volume, *V*, of the spherical liquid cap is given by
(8)$$V=\frac{\pi R^{3}}{3}{\left(1-\cos\;\theta \right)}^{2}\left(2+\cos\;\theta \right)\;\to \mathrm{or}\;V=\frac{\pi R^{3}}{3}{\left(1-u\right)}^{2}\left(2+u\right)$$

Expressing Equation (8) in terms of *R* yields
(9)$$R^{3}=\left(\frac{3V}{\pi }\right){\left(1-u\right)}^{-2}{\left(2+u\right)}^{-1}$$
and therefore,
(10)R2=3Vπ231−u−432+u−23

Equilibrium is achieved when *G* is minimized with respect to *u*. Taking the derivative of *G* with respect to *u*,
(11)$$\frac{\partial G}{\partial u}=-\gamma_{LV}2\pi R^{2}-(\gamma_{SL}-\gamma_{SV})2\pi R^{2}u+\left[\gamma_{LV}2\pi \left(1-u\right)+(\gamma_{SL}-\gamma_{SV})\pi \left(1-u^{2}\right)\right]\frac{\partial R^{2}}{\partial u}\;$$

Taking the derivative of Equation (10) and noting that *V* remains constant yields
(12)∂R2∂uV=3Vπ23431−u−732+u−23−231−u−432+u−53

Rearranging Equation (12) yields
(13)∂R2∂uV=3Vπ231−u−432+u−23431−u−1−232+u−1

After substituting for $R^{2}$ (see Equation (10)) into Equation (13) and simplifying, we obtain
(14)$${\left.\frac{\partial R^{2}}{\partial u}\right|}_{V}=R^{2}\left[\frac{4}{3}{\left(1-u\right)}^{-1}-\frac{2}{3}{\left(2+u\right)}^{-1}\right]=2R^{2}\left[\frac{\left(1+u\right)}{\left(1-u\right)\left(2+u\right)}\right]$$

Substituting Equation (14) into Equation (11) and setting ${\left.\frac{\partial G}{\partial u}\right|}_{V}=0$ (i.e., *G* is a minimum at equilibrium), yields
(15)$${\left.\frac{\partial G}{\partial u}\right|}_{V}=-2\pi R^{2}\left(\gamma_{LV}+(\gamma_{SL}-\gamma_{SV})u\right)+\frac{2\pi R^{2}\left(1+u\right)\left(2\gamma_{LV}+(\gamma_{SL}-\gamma_{SV})\left(1+u\right)\right)}{\left(2+u\right)}=0$$

Simplifying Equation (15) and back substituting cosθ for u recovers Young’s equation,
(1)$$\gamma_{SV}=\gamma_{SL}+\gamma_{LV}\cos\;\theta ,$$

When *θ* is between 10° and 90°, a surface is considered to be hydrophilic; a value between 90° and 150° indicates that the surface is hydrophobic. A surface is regarded as superhydrophilic or superhydrophobic when *θ* is less than 10° or more than 150°, respectively. Using typical values for the interfacial energies associated with a water drop on a smooth hydrophobic surface [25] (for e.g., a close packed monolayer of hydrocarbons with CH_3_ terminal groups) in air at 22 °C, i.e., $\gamma_{SL}=50$ mN/m,$\;\gamma_{SV}=18.5$ mN/m, and $\gamma_{LV}=72.5$ mN/m, a WCA of 116° is obtained according to Young’s Equation (1). The latter provides a rough estimate for the maximum water contact angle on a typical smooth, homogeneous hydrophobic surface.

Real surfaces are not perfectly smooth, as idealized by Young’s Equation (1). For rough surfaces, the Wenzel model or the Cassie–Baxter model can be used to predict the contact angle of a droplet over the surface. According to Wenzel’s model [26], the water droplet penetrates the grooves of a rough surface and completely wets it (Figure 5a). The contact angle of a rough homogeneous surface is given by Wenzel’s equation,
(16)$$\cos\theta_{w}=r\cos\theta ,$$
where $\theta_{w}$ is the apparent Wenzel contact angle, θ is the static contact angle and *r* is the roughness factor (which is >1) given by the ratio of the actual surface area to the geometric surface area. The value of *r* is equal to 1 for perfectly smooth surfaces, in which case Wenzel’s Equation (16) will be equivalent to Young’s Equation (1). For a rough hydrophilic surface (i.e., θ < 90°), $\theta_{w}$ < θ and $\theta_{w}$ > θ for a rough hydrophobic surface (i.e., θ > 90°). Note that for a contact angle θ = 90°, according to Wenzel’s Equation (16), the roughness factor does not influence the contact angle.

The Wenzel model becomes inadequate in describing very rough or porous surfaces since the absolute value of rcosθ may be greater than 1. In such instances, the Cassie–Baxter model [27] is more applicable. In this model, wetting occurs on a porous heterogeneous surface as shown in Figure 5B. The water droplet does not penetrate the grooves but rests on the top of asperities thus maintaining air pockets between the solid phase and the liquid phase. The apparent contact angle is given by
(17)$$\cos\theta_{CB}=f_{1}\cos\theta_{1}+f_{2}\cos\theta_{2}$$
where $\theta_{CB}$ is the apparent Cassie–Baxter contact angle, and $f_{1}$, $f_{2}$ and $\theta_{1}$, $\theta_{2}$ are the surfaces fractions and contact angles of phases 1 (solid) and 2 (vapor), respectively. Since the contact angle that a water drop makes with air, i.e., $\theta_{2}$, equals to 180° and since $f_{1}$ + $f_{2}\;$= 1, Equation (17) simplifies to
(18)$$\cos\theta_{CB}=f_{1}(\cos\theta_{1}+1)-1,$$

For a perfectly smooth surface, $f_{1}$ = 1 and $\theta_{CB}=\theta_{1}$. However, $f_{1}$ is typically less than 1, thereby causing $\theta_{CB}>\theta_{1}$ for either hydrophilic or hydrophobic surfaces. The lotus leaf effect is a perfect example of the implementation of the Cassie–Baxter model.

The three models explained above describe hydrophobicity for static conditions where the surface is horizontal and at rest. The term ‘water repellency’, albeit used interchangeably with hydrophobicity, is more frequently used to describe non-static (or dynamic) situations. A surface can be hydrophobic or superhydrophobic yet, when tilted at an angle, a water droplet can remain adhered to the surface, i.e., does not slide. While static contact angles are used to describe hydrophobicity (or hydrophilicity), the contact angle hysteresis (CAH), which is the difference between the advancing and receding contact angle (i.e., $\theta_{A}$ and $\theta_{R}$), defines water repellency (see Figure 6). CAH is a widespread phenomenon which is believed to originate from nonequilibrium interactions between a liquid and a solid [25]. Although its origin is not fully understood, it has been found to depend on several factors including the rate of movement of the 3-phase contact line, surface roughness, the adsorption of reagents on the surface, the strength of adhesion of water and air molecules to the solid surface [28], and the overturning of adsorbed molecules on the surface [29]. The force, *F*, required to initiate the movement of a liquid drop over a titled solid surface [30] is related to the CAH and given by(19)$$F=\gamma_{LV}Rk\left(\cos\theta_{R}-\cos\theta_{A}\right)\;$$
where *R* is the radius of the drop and *k* is a numerical constant that depends on the shape of the drop. This force is also equal to $mg\sin\theta_{sliding}$, i.e., the force component originating from the mass, *m*, of the liquid drop due to gravity, *g*, that overcomes the opposing friction force of the drop. Based on the above equation, as the CAH decreases, this critical line force decreases. Therefore, effective water repellent surfaces have very low CAH values. The term “superhydrophobicity” refers to a surface not only having very high WCA (i.e., greater than 150°) but also having a very low CAH which results in a low SA.

## 3. Micro- and Nanofabrication Techniques to Create Superhydrophobic Surfaces

Over the last two decades, researchers have explored a variety of methods to generate SH surfaces on various materials for a wide range of applications. SH surfaces are typically fabricated in one of two ways: (i) creating roughness on an inherently hydrophobic surface; or (ii) modifying a rough surface with a low surface energy coating. Most fabrication methods rely on a multi-step process while others require a single step. In multi-step methods, micro/nanostructures are initially fabricated on the surface followed by modification of the surface by coating or deposition techniques. Multi-step fabrication methods are usually complicated, costly, environmentally hazardous, and inefficient [31]. On the other hand, one step methods are generally fast and can easily be applied to large-scale industrial manufacturing [32].

In this review paper, some of the main fabrication methods are discussed with particular emphasis on and brief description of techniques that have received the greatest interest. This section first introduces coating and deposition techniques followed by chemical-based techniques and finally ends with physical-based techniques. A summary of the advantages and disadvantages of each of the fabrication techniques is also provided. In Figure 7, an overview of different fabrication techniques of SH surfaces is provided categorized as either physical or chemical techniques, or a combination. It is worth noting that in techniques such as lithography and laser surface texturing, the dominant part of the process is physical. However, coating techniques can be considered to be a combination of both physical and chemical since chemical cross-linkers can be used to improve the durability of the deposited films.

### 3.1. Surface Coating and Deposition Techniques

Coating a surface with nano/microparticles is one of the most reliable techniques to generate structures/roughness on surfaces. To make the coating stable, surface treatments are required which create functional groups for the formation of strong covalent bonds between the surface and the coating [33]. Physical deposition techniques like dip-, spray-, and spin-coating are described below. These techniques are relative easy, cost-efficient, and scalable, which make them good candidates for industrial applications [34].

#### 3.1.1. Spin Coating

Using a spin-coater, a solution with low surface energy can be coated on the substrate. Xu et al. [35]. used this technique to deposit a layer of epoxy resin on an aluminum substrate thereby imbuing anti-corrosion properties. Dendritic micro/nano structures were formed on a copper mesh, and then transferred to the coated aluminum surface. It was then modified with stearic acid to reduce the surface energy (Figure 8). Due to the presence of the organic epoxy coating and air gaps, the fabricated surface showed considerable anti-corrosion performance compared to bare aluminum. The self-cleaning performance and repellency of the surfaces against acids and bases showed promising results.

Generally, spin-coating is not a single-step fabrication method, and a post treatment step is needed to further lower the surface energy. Moon et al. [36] spin-coated a polydimethylsiloxane-polyvinylidene fluoride (PDMS-PVDF) solution on stainless steel (SS), and SiO_2_ powder was sprinkled using a mesh to coat the surface uniformly. The surface was then cured at 160 °C, and the unreacted SiO_2_ was removed. Another surface was prepared with no addition of PVDF to compare the characteristics. It was found that PVDF improved the mechanical properties of the surface by providing nanostructures between the mixture and SiO_2_ powder. Also, the presence of PVDF improved the icephobicity which was ascribed to the formation of the non-polar α-phase crystals of PVDF on the surface of SiO_2_ particles.

In another study, Cho et al. [37] reported spin-, dip-, or spray-coating of a hydrophobic silica ormosil suspension and poly(methylhydroxysiloxane) (PMHOS) nanoparticles on a substrate to increase the surface roughness. The surface was then chemically modified with self-assembled monolayers (SAMs) of silane-based molecules (n-alkylsilane, phenylsilane and fluorosilane) to further reduce the surface energy and make the surface superamphiphobic. The surface modified with fluorosilane showed the lowest energy. The contact angles of water, ethylene glycol, and diiodomethane on the F-ormosil/PMHOS surface were 174°, 157°, and 131°, respectively.

#### 3.1.2. Spray Coating

In the spray-coating technique, hydrophobic particles are dispersed in a solvent and then sprayed on a substrate [15]. The solvent and the spraying distance are two of the most important factors that affect the water repellency and robustness of the fabricated SH surfaces [34]. The adhesion of the coating to the substrate is also critical [38]. Researchers have used strong adhesives such as an epoxy resin, functionalized nanoparticles, and polyurethane (PU) to enhance the adhesion. PDMS has also been used since it not only enhances the adhesion, but also, because PDMS is inherently hydrophobic, such a coating is favorable in reducing the surface energy [39,40,41,42,43,44,45,46].

Li et al. [44] spray-coated a fluorine-free suspension (epoxy resin, PDMS, and functionalized SiO_2_ nanoparticles) onto various substrates, including Mg alloys, glass, filter paper, copper foam, and sponge, to fabricate robust SH coatings (Figure 9a–l). Superhydrophobic SiO_2_ nanoparticles formed mechanically stable micro-nanostructures due to the presence of epoxy resin layer. The as-prepared composite exhibited durability against harsh conditions, including boiling water, intense UV light exposure, and chemical corrosion. The WCA and SA of the surface were 159.5° and 3.8°, respectively. Zhang et al. [43] used spray coating of a mixture of micro-nano SiO_2_ particles, epoxy resin, and PDMS on various substrates (including glass, filter paper, copper sheets, and polyethylene terephthalate (PET) films (Figure 9m). The as-fabricated SH surfaces showed high chemical and mechanical durability due to strong bonds between epoxy and PDMS [43].

Feng et al. [47] used spray coating to deposit polysiloxane-modified halloysite nanotubes (POS@HNTs) in a toluene suspension, with different ratios of silane and HNTs, on various substrates and the as-fabricated surfaces showed high repellency towards various liquids, including hydrochloric acid, aqueous sodium hydroxide, tea, and milk. It was shown that by increasing the silane loading, although the transparency of the surface decreased, the WCA increased. In another study, Zhou et al. [48] first spray-coated PDMS (as the binding layer) and then a solution of silicone nanofilaments (NF) in toluene on a pre-stretched substrate (a commercial cis-1,4-polyisoprene tape), followed by curing the surface at 80 °C for 2 h, and fluorination by chemical vapor deposition (Figure 10). The NFs formed dense re-entrant nano- and micro-scale structures that showed repellency against water and oil (n-hexadecane) for at least one thousand stretch-release cycles, resulting in a durable stretchable superamphiphobic surface.

Specifically for oil–water separation technologies, it is desirable to have membranes that are superhydrophobic yet superoleophilic [46,49]. Baig et al. [49] proposed spray coating cerium (IV) oxide nanoparticles on microporous SS membranes which yielded a WCA of about 150° and an oil contact angle close to 0°. The surface showed a high oil–water separation efficiency of about 99%. In addition, under exposure to UV light, the as-prepared membrane exhibited fast photocatalytic degradation of hazardous environmental pollutants such as organic dyes in the water phase.

#### 3.1.3. Dip Coating

Self-cleaning (Figure 11F) [50], self-healing [51], anti-corrosion characteristics [52], and oil–water separation [53] are some of the main properties achieved by using a dip-coating technique. The dip-coating technique typically has five steps, including dipping, start-up, deposition, drainage, and evaporation [21]. When the substrate is pulled out, the speed can be controlled to dictate the thickness of the film [54]. Dip-coating is generally not an environmentally friendly technique because of the chemicals used as the solvent. However, some groups have proposed non-toxic solvents, for e.g., fluorine-free chemicals. Esmaeili et al. [55] reported dip-coating substrates (glass and paper) in non-toxic molten alkyl ketene dimer (AKD) for a few seconds and removing instantly. After solidification of the AKD on the substrates at room temperature, some of the samples were wetted by ethanol and dried in air. Further treatment with ethanol, and increased temperature exposure and heating time resulted in more entangled irregular micro/nano structures, leading to more air gaps and increased superhydrophobicity.

Sosa et al. [32] described a facile one-step scalable method to fabricate mechanically and chemically stable SH coatings on metal meshes (Figure 11A–E) for oil–water separation. The process consisted of coating inorganic hydrophilic SiO_2_ nanoparticles (to provide nano-scale roughness) and two hydrophobic polymers (PDMS and PVDF) in toluene, followed by curing at 200 °C for 2 h. The fabricated membranes provided a WCA of 158 ± 6°, SA of 3 ± 2°, and a stable oil flux on the order of about 10^2^ Lm^−2^s^−1^. Saline (Na^+^, Cl^−^) solutions or hard water (Mg^2+^, Ca^2+^) did not change the morphology or wettability of the membranes because the SiO_2_ nanoparticles were coated with PDMS, thereby preventing contact between the aqueous solutions.

In conclusion, spin-, dip-, and spray-coating are simple and cost-efficient techniques because of the relatively low temperature and pressure requirements, and therefore ideal for large-scale industrial applications. They also provide high-quality, uniform films on surfaces of various substrates (even complex ones) within a short time. However, the fabricated surfaces are not always stable unless a cross-linking agent (like PDMS and epoxy resin) is used. In some cases, the coatings can show long-term durability with self-healing properties upon exposure to an external stimulus such as heat, sunlight, and pH change [44,56,57,58]. One of the major drawbacks to the dip-coating technique is the amount of solvent needed, causing considerable waste. Depending on the scale and compatibility of the substrate to the coating method, spray-coating can be easier and more efficient than spin- or dip-coating [21,54,59,60,61]. Furthermore, using the spray-coating technique, if an SH surface is damaged (for e.g., because of surface abrasion), it can be repaired by local spraying of the coating solution.

### 3.2. Chemical Etching

Chemical etching is the most common type of etching method due to its simplicity, scalability, and low cost. The technique generally involves immersing an alloy into solutions such as HCl [62,63], MgCl_2_ [64], HNO_3_ [65], FeCl_3_ [66] followed by a modification step using a fluoroalkylsilane [64,65,67,68]. The etching time, the components and their concentrations in the solution can endow the substrate with different morphologies (Figure 12). Guo et al. [64] fabricated a 7055 aluminum alloy with micro- and nano-hierarchical structures. The substrate was ultrasonically cleaned and electropolished to remove the oxide layer. Then, using a hydrothermal reactor at a temperature of 90 °C, the substrate was immersed in a 0.7 g/L MgCl_2_ solution to create the hierarchical micro-nanotextures. Due to the presence of chloride ions, Al^3+^ and H^+^ were formed at the interface of liquid/surface, leading to the reaction ${\mathrm{Al}}^{3+}+3H_{2}O\to \;\mathrm{Al}{\left(\mathrm{OH}\right)}_{3}+3H^{+}$, thereby forming a uniform Al(OH)_3_ coating. Subsequently, in order to reduce the surface energy of the etched substrate, it was modified by immersion in a 1 vol.% perfluorooctyltriethoxysilane/anhydrous alcohol solution for 10 min to achieve a WCA of 167.3° and a corrosion inhibition efficiency of 99.67%.

The drawback of this method is the use of acidic and toxic solutions that cause environmental issues, especially on an industrial scale. In the modification stage, using a non-toxic material including PDMS [62] and stearic acid [63], instead of fluorine based components is recommended to lessen the environmental impact. Xu et al. [69] fabricated SH PDMS films through chemical etching and thermal curing processes. First, nickel templates with micro-pyramid arrays were chemically etched using nitric acid to increase the surface roughness and create air gaps. Densely distributed protuberances and mastoids roughness were achieved on the micro-pyramids, using 45% and 55% HNO_3_ solutions, respectively. Thereafter, PDMS was poured on the etched surfaces and thermally cured at 120 °C for 6 min. After peeling the cured PDMS from the templates, a WCA of 166.4° ± 0.5° and a SA of 9.6° ± 0.2° were obtained on the PDMS film with mastoid textures on pyramids.

### 3.3. Sol–Gel Process

The sol–gel process is a wet-chemical technique that generally comprises of two steps. The first step is the hydrolysis of nanoparticles such as SiO_2_ [70,71,72,73], TiO_2_ [74,75], ZnO [76,77], in a solvent to produce a sol, followed by polycondensation reactions of the hydrolyzed product to form a gel. Either in the condensation or the post-treatment step, particles are usually modified by non-fluorinated silica based materials, for instance, PDMS [78], 1,1,1,3,3,3-hexamethyl disilazane (HMDS) [74], and tetraethyl orthosilicate (TEOS) [79,80]. The main aim of the condensation step is to maximize −Si–O–Si− bonds and minimize Si–OH and Si–OR groups [81]. Using the sol–gel process, it is possible to combine multiple particles to take the advantage of their properties simultaneously. For example, a coating of SiO_2_/TiO_2_ composite shows photocatalytic properties due to the presence of TiO_2_ and enhanced adhesion, durability, and hydrophobicity because of SiO_2_ particles [82].

SH coatings produced by this technique can be applied to various substrates (metal, glass, wood, polymer, and fabric) by dip coating, spin coating, spray coating, or electrodeposition (Figure 13b). Zhu et al. [80] fabricated SH ferric oxide nanoparticles coated with HMDS-modified silica particles via a one-pot sol–gel technique. Based on the modified Stöber method [83], TEOS and HMDS were added to a mixture of Fe_3_O_4_ nanoparticles, DI water, NH_3_·H_2_O, and ethanol, under sonication (Figure 13a). The modified Fe_3_O_4_ particles were added to a PDMS mixture to enhance the hydrophobicity and the adhesion strength with the substrates. Using a simple spraying method, the SH particles were coated on different fabrics to endow the substrates with durable water repellency. The coated fabrics showed superhydrophobicity even after 200 abrasion cycles and had an oil/water separation efficiency greater than 97%.

Xia et al. [23] reported a method to fabricate micro/nanostructured SiO_2_ (SH coating) on a wood surface by a sol–gel process. CVD was used to modify the wood surface with a low surface energy substance, 1H, 1H, 2H, 2H-perfluorodecyltrichlorosilane (PFDS). The presence of an unstable Si-Cl in the PFDS molecule, silanized the surface and favored the production of the SH coating under mild conditions. The modified wood not only possessed SH properties (WCA of 159° and SA of 6°) but also had significantly enhanced physical properties including thermal stability, acid and base resistance, and abrasion resistance.

The sol–gel technique has numerous advantages, including ease of manufacture and controllable compositions [85], eco-friendliness and low toxicity to human health. It is a convenient and versatile way to produce fine microstructures at low temperatures [84]. It is also the most efficient way to produce crystalline or amorphous oxide coatings and complex shapes with controllable surface properties without the use of specialized instruments [86]. However, a disadvantage of the sol–gel technique is that the thickness of the coating is not easily controlled, and contractions can occur during processing [87]. Hu et al. [88] discovered that when the substrate is a flexible polymer, the interfacial bonding between the polymer and the inorganic particles via conventional sol–gel methods is usually poor, making the coating highly susceptible to damage. To address this problem, cured silicone rubber (SR) was swollen in advance using an aqueous solution of n-butylamine (sol–gel catalyst) and was then immersed in a tetrabutyl titanate (TBT) solution to facilitate the sol–gel reaction. The precursor diffused into the SR and combined with the catalyst, leading to the formation of TiO_2_ particles in the crosslinking network of the SR. The TiO_2_ particles gradually grew and formed a texture with multiscale roughness on the SR surface. The self-growth of the TiO_2_ particles in the crosslinking network of the SR ensured that the TiO_2_ endowed the surface with mechanical durability [88].

### 3.4. Electrodeposition

Electrodeposition (ED) is a low-cost and scalable method in which an electric potential is used for the reduction of metal ions and subsequent deposition on a substrate acting as a cathode. It can be applied to various metals, including aluminum, magnesium, steel, and copper. However, the process requires that the substrate is electrically conductive. The main advantage of this method is the control of morphology by adjusting the choice of the cathode and parameters such as the current density, voltage, electrolyte composition, temperature, and pH [89,90]. For instance, by anodization of copper and Al, hydroxide nanoneedles and cylindrical nanopores close-packed in a hexagonal shape can be obtained, respectively [91]. Crystal modifiers including ethylenediamine dihydrochloride (EDA.2HCl) [92] and ethylenediammonium dichloride (C_2_H_10_Cl_2_N_2_) [93] endow magnesium alloys and copper substrates with pinecone-like structures, respectively. In order to reduce the surface energy of structured substrate, the latter can be modified either in the ED step or in the post-treatment step. Stearic acid has been a common component for this modification as a non-toxic, biocompatible, and low surface energy saturated fatty acid [94,95,96]. Liu et al. [95] fabricated a durable stearic acid/CeO_2_ SH coating on AZ31B Mg, through electrodeposition and modification in a stearic acid solution. The as-prepared surface displayed WCA of more than 158° and SA of less than 2°.

To enhance the durability and stability of surfaces, other types of ED processes including electrophoretic deposition (EPD) and electropolymerization (EP) have been studied. EP involves the deposition of oxidized monomers on the substrate. Fradin et al. [97] fabricated nanotubes on a gold plate through the EP process. The electrolyte contained tetrabutylammonium perchlorate (Bu_4_NClO_4_), carbazole-based monomers (with various molecular structures), and dichloromethane saturated with water (CH_2_Cl_2_ + H_2_O). They showed that the morphology of sample a is not only a function of EP operational parameters but also highly dependent on the water concentration in the electrolyte and the molecular structures of the monomers.

EPD involves the deposition of charged particles from a colloidal dispersion. The deposition of particles on the surface can provide roughness and enhanced durability due to the electrostatic interaction between the substrate and the coating. Since the uniformity of the coating is essential for its stability, using nanoparticles [90] in a homogenous mixture is recommended. Through the EPD process, an SH surface can be created in a single step which is desirable for industrial production scales. Zhang et al. [98] fabricated a SH surface through a one-step EPD process using dodecyltrimethoxysilane (DTMS) modified SiO_2_ nanoparticles and an acrylic-based resin adhesive (Figure 14). The role of resin was to act as a cross-linker to enhance the interactions between the particles and the surface. For further corrosion improvement, benzotriazole (BTA) was added to the electrolyte. When the surface is damaged and corrosive compounds penetrate the substrate, changes in pH activates the BTA inhibitors to slow down the corrosion reaction thereby generating physical barriers to increase the lifetime of the surface.

### 3.5. Chemical Vapor Deposition

In the chemical vapor deposition (CVD) method, a substrate is exposed to a gaseous phase of volatile precursors which react to form a film or powder layer on the solid substrate. Owing to the covalent bonds between the film and the substrate, the fabricated SH surface is typically mechanically stable. It is a suitable technique for modifying and growing nanomaterials with high specific surface ratios [87]. The thickness, uniformity, and morphology of the film are influenced by the treatment temperature, pressure, gas flow, and the deposition time. Generally, the temperature and deposition time are in the range of 230 °C–450 °C and 30 min–16 h, respectively [99,100]. Since the CVD process generally involves high operational temperatures, pressures and expensive instruments, the process is not cost-efficient, and not practical for large-scale applications [87].

Hence, plasma enhanced CVD (PECVD) and photo-induced CVD (PICVD) have been proposed as modified versions of the conventional CVD process to reduce the temperature and pressure requirements. PECVD uses the energy of electrons to initiate the heterogeneous reactions producing active ions and radicals to deposit the coating under ambient conditions [101]. Montes et al. [102] grew 3D micro and nanostructures of TiO_2_ on flat or porous substrates via PECVD. In the first step, TiO_2_ nanofibers (as nucleation centers) were deposited through PECVD at room temperature and under 100% oxygen plasma. Figure 15a,b show SEM images of the cauliflower morphology obtained by the PECVD in step 1. Thereafter, through a simple physical vapor deposition (PVD), H_2_-phthalocyanine (H_2_Pc) nanowires (core) were created on the TiO_2_ layer (shell) to fabricate a core@shell morphology with multiple scales. At this step, the morphology is a function of growth rate (Figure 15c,d), substrate temperature, deposition time, and the size of nucleation sites. In step 3, an organic and inorganic shell were deposited through another PECVD to achieve the thickness of 200 nm, which was optimized by the deposition time. Finally, the coated substrates were modified by fluorine-based grafting and PDMS respectively to endow the substrates with superomniphobicity that can separate fluids based on their surface tensions and phases. Given the use of plasma in PECVD, it is the most energy-intensive technique among all CVD methods. However, such high energy can not only damage fragile substrates [103] but also weaken cross-linking [104].

PICVD consists of the interaction between light radiation, precursors molecules, and substrates either in the gas phase or on the substrate. Instead of thermal energy, absorbed energy from excimer lamps enables the precursors molecules to break chemical bonds. Like PECVD, PICVD involves ambient temperatures and mild pressures. Furthermore, compared to other CVD methods, the deposition rate and cross-linking are sufficiently high to meet industrial standards. Therefore, PICVD can be an efficient and promising technique for large-scale applications [101].

Combining the CVD process with other fabrication techniques can have the added benefits of reducing the required operating temperatures and also improving the durability of a SH coating. Barthwal et al. [62] modified an etched and hydrothermally treated Al surface via conventional CVD. PDMS was used as a cost-efficient, chemically resistant, nontoxic, and low surface energy material to coat the top layer. The etched surface was placed horizontally 5 cm above the PDMS stamp to be exposed to the PDMS at 230 °C. After 10 min, the surface achieved superhydrophobicity with a 10-month durability.

Cheng et al. [105] described an enzyme-etching method followed by modification with methyltrichlorosilane (MTCS) via thermal CVD at 70 °C to fabricate SH silk fabrics. The silk fabrics transformed from superhydrophilic to SH after enzymatic treatment (through the hydrolysis of the fabric via the enzyme) and MTCS modification (Figure 15e). This transformation was ascribed to the increased roughness caused by enzyme-etching and low surface energy of the methyl groups of MTCS. With optimized experimental conditions, MTCS@enzyme-etched fabrics displayed a WCA of 156.7° and a SA of 8.5°. This technique can potentially be applied to a wide range of fabrics through the judicious choice of the enzyme.

### 3.6. Plasma Processing

Plasma processes encompass the various technologies in which plasma species (ions, radicals, electrons, and excited molecules) are used to develop nanostructures on the surface of substrates and reduce surface energy. There are two approaches to synthesize SH surfaces by plasma treatments. The first is the top-down approach or plasma etching which consists of etching or partially removing material from the surface of the substrate by plasma species, followed by grafting with non-wetting radicals to lower the surface energy [106]. Plasma etching results from the interactions between the plasma and the surface of the substrate, using particles with high kinetic energy to knock out the atomic or molecular species from the substrate.

An advantage of plasma etching is that it allows controlled and precise etching down to the nanometer scale. In addition, plasma etching can produce high aspect ratio features. Ko et al. [107] found that plasma-based selective etching can be used to fabricate SH surfaces with high aspect ratio nanostructures. They proposed a one-step process to fabricate hierarchically structured SH surfaces with tunable water adhesion by plasma-based selective ion etching supported by the dual scale etching mask. The metallic mesh had microscale walls and nanoscale high-aspect-ratio structures with various gap distances (Figure 16a–d). The hierarchical textured surface consisted of nanostructures in a large area with a larger gap distance, however, the hybrid surface included nanostructures nested in microscale walls with shorter gaps (Figure 16e,f). Superhydrophobicity was achieved on both surfaces, but the adhesion force on the hierarchical surface was 10 times less than the hybrid surface. The main disadvantages of plasma etching are low selectivity and high equipment costs.

The second approach is the bottom-up technique called plasma enhanced chemical vapor deposition (PECVD), which involves the growth of micro or nano structures on the substrate (refer to Section 3.5).

### 3.7. Laser Surface Texturing

There are three main laser texturing techniques: Direct Laser Writing (DLW), Direct Laser Interference Patterning (DLIP), and Laser-Induced Periodic Surface Structuring (LIPSS) which generates structures within the scale of 10 to a few hundred micrometers, 1 to 10 μm, and less than 1 μm, respectively. Based on the size scale of the desired structures the appropriate technique can be selected [108]. During laser surface texturing (LST) by direct laser ablation, a laser beam is irradiated on the surface, resulting in the localized evaporation (and loss) of the material and formation of specific surface textures, such as grooves. DLIP uses coherent laser beams that interfere with each other, again resulting in localized melting and transfer of material from regions with maximum interference (i.e., highest laser intensity) to regions with minimum interference because of the surface tension gradient generated. Various periodic patterns such as lines and dots can be formed by this technique [109]. During LIPSS, the material (including semi-conductors, metals, and dielectrics) is exposed to one or multi-pulses of laser with energies higher than the modification threshold. LIPSS involves interference of an incident laser radiation and the waves scattered from the irradiated material of the surface [108,110]. In addition to periodic structures formed by the laser, self-organized conical micro/nanostructures can be induced by repetitive exposure of ultrafast laser pulses (mainly femtosecond) on the surface when the laser fluence is equal to or higher than the ablation threshold [111,112,113,114].

LST can be used on a variety of substrates, including semiconductors, metals, polymers, glasses and ceramics, making the technique suitable for a wide range of applications. To fabricate SH surfaces, one-step laser processing [115], the combination of laser techniques [116,117], and laser techniques followed by surface treatments [114,118,119] have been explored by various researchers. Milles et al. [117] used DLW and DLIP, and a combination of these two technologies to fabricate SH structures on aluminum surfaces (Figure 17). The DLW and DLIP resulted in mesh-like and pillar-like structures, respectively. After 16 days, the laser treated surfaces attained a SH state.

Different process parameters including scanning distance, laser fluence, pulse repetition rate influence the size and quality of the resulting structures [112,115,120,121,122]. Lu et al. [123] investigated the effect of laser fluence (2.69–9.55 J/cm^2^) on fabricating different sizes of micro-cracks and microstructures on 316L SS substrates by UV nanosecond laser texturing. It was shown that microcracks were created on the surface at laser fluences lower than 8.14 J/cm^2^, and the cracks became deeper by increasing the laser fluence. However, at laser fluences higher than 8.14 J/cm, brain-like microstructures were formed, creating air-gaps resulting in superhydrophobicity (Figure 18). A scanning interval of 30.0 μm and laser fluence of 8.14 J/cm^2^ showed the best superhydrophobicity with WCA of 160 ± 5° and the SA of 3 ± 0.5° and self-cleaning properties. The corrosion resistance of the fabricated SH surface was 88 times superior compared to bare 316L SS. Du et al. [122] studied the effect of laser power intensity and pulse overlap on the wettability of polyimide surfaces fabricated through UV laser direct texturing. It was found that at constant pulse overlaps, increasing the laser power intensity caused an increase in the WCA. At low power intensities (<5.5 × 105 W/cm^2^), increasing pulse overlaps transformed the hydrophilic surface to a SH surface. However, at higher power intensities, increasing the pulse overlaps resulted in superhydrophilicity.

The effect of scan spacing was investigated by Lian et al. [31] where dual- and three-level structures were fabricated on an aluminum alloy plate by a nanosecond laser processing system (Figure 19). The nanosecond laser oxidized the Al materials and produced an oxide film on the surface. The surfaces were then placed in an oven for low-temperature (100 °C) annealing to change the surface chemical composition. After the two-step procedure, the surfaces were found to be SH, showed anisotropic wetting properties and corrosion resistance.

There are various laser sources with different pulse durations, including femtoseconds [113,124], picoseconds [118,119], nanoseconds [115,125], and long pulse times [108]. Nanosecond pulse lasers are more cost and time-efficient compared to femtosecond and picosecond pulse lasers but can cause thermal effects. By using nanosecond pulse lasers, heat affected zones (HAZs) are created because of thermal energy diffusion to the areas around the irradiated spot. HAZ can lead to microcrack formation and affect material integrity. In addition, using longer pulses can result in the formation of re-cast layers, thereby lowering the resolution of structures. On the other hand, shorter pulse lasers such as femtosecond lasers provide higher precision and minimal thermal effects around the ablation spot. The main disadvantage of shorter lasers is that they tend to be more expensive [120,121,126,127]. Khan et al. [121] investigated the wettability and self-cleaning performance of three laser ablated metals (aluminum, copper and galvanized steel) with femtosecond and picosecond pulse durations with different scanning speeds, followed by storage in air and in a low-pressure environment. At a scanning speed of 1000 mm/s, femtosecond lasers created more elevated and precise micro/nanostructures compared to nanosecond laser, because of rapid vaporization and removal of material in the lower interaction time with the metal. Using a low scanning speed (50 and 250 mm/s), aluminum showed hydrophilicity after picosecond laser ablation, and hydrophobicity after femtosecond laser ablation. The difference in wettability is related to the electron-phonon relaxation time which is in the range of picoseconds; therefore, different laser pulses result in different mechanisms for material ejection. However, copper and galvanized steel showed similar wettability after being ablated with femtosecond or picosecond laser, at low or high scanning speed, because these laser pulses are considered short compared to electron-phonon relaxation time of these metals. It was also found that the surfaces transformed to the hydrophobic state after 30 days under ambient conditions, and to the SH state after only 6 h in a low-pressure environment.

Sun et al. [118] reported a picosecond laser patterning to fabricate micro/nanostructures on AISI 304 SS, followed by spin coating the surface with a silicone sol (30 wt.% spherical silica nanoparticles (NPs)). A heat treatment step was used to initiate a polycondensation reaction and the formation of Si-CH_3_ on the NPs, resulting in a lower surface energy and improved adhesion between the substrate and the NPs. The micro-groove patterned surface was found to reduce microbe attachment area rate (MAAR) by about 50% after five weeks immersion in seawater making the textured SH surface also antibiofouling.

Laser processing is an efficient, facile, and non-contact method for fabricating various complex structures with high precision [123]. Compared to conventional methods (like spray coating, dip coating and chemical etching), laser processing techniques are faster, and the fabricated structures are more stable [60]. Generally, sample preparation, harsh environment, photomasks and hazardous chemicals are not required in laser processing as opposed to other techniques. Not using chemicals make this technique relatively environmentally friendly. However, this technique can be expensive because of the equipment cost (especially in the case of femtosecond lasers) and generally, not ideal for large-scale surface texturing [38,108].

### 3.8. Electrospraying and Electrospinning

Electrospinning and electrospraying both use applied electric fields between an injector and a conductive substate to deposit nano/microstructures from polymer solutions, melts, and viscoelastic liquids [128]. The set-up and operating principle behind these two techniques are similar, consisting of a feeding system with a container that holds the liquid precursor, a spinneret and a pump that injects the precursor at a constant speed. A high voltage is applied to generate a sufficiently strong electric field to overcome the surface tension of the solution thus allowing the melt to be ejected onto the grounded collector (Figure 20).

A fundamental difference between electrospinning and electrospraying is based on the concentration of the polymer solution. Low concentrations are used in electrospraying, while electrospinning processes use high concentration solutions/melts. The concentration of the solution is critical to the morphology. At low concentrations, spherical particles are obtained. As the concentration increases, bead-on-string morphologies are typically obtained. If the concentration is above a critical value, uniform fibers are formed.

Cheng et al. [129] proposed a coaxial electrospinning method to fabricate a SH membrane for ultrafast oil/water separation. The hierarchical and micro/nanoscale morphology was obtained by using a PVDF solution as the lumen solution and a silica-based solution as the outer solution under a voltage of 22 V (Figure 20a). Solution A (lumen solution) contained 10%wt PVDF with dimethylformamide (DMF) and tetrahydrofuran (THF) as solvents. Solution B (outer solution) contained 10%wt PDMS and THF as the solvent. As the solutions were ejected from the syringes, phase inversion occurred with PDMS monomers cross-linking. Owing to the polymeric viscosity differences (PVDF and PDMS) and cosolvents evaporation (THF and DMF), the final structure consisted of microspheres attached to nanofibers (Figure 20c). The nanofibrous SH membrane provided a permeance of 17,331 Lm^−2^h^−1^ for separating a water-in-n-octane mixture with an efficiency of 99.6%.

The advantages of electrospinning include the simplicity and efficiency of the process, low-cost setup, and the ability to control many factors such as fiber diameter, orientation, and composition. The limited control of pore structures and the use of organic solvents are some of the disadvantages of this method. It has been suggested that a needleless electrospinning technique has a higher production rate and can be equipped for large-scale production of fiber mats [131]. Like electrospinning, electrospraying has advantages including simplicity and ease in controlling operational parameters. As a disadvantage, this technique can induce some macromolecular degradation because of the shear stress in the nozzle and thermal stress during drying [132].

### 3.9. Lithography Patterning Technique

Lithography can be sub-categorized into photolithography, X-ray lithography, electron beam lithography, colloidal lithography, soft lithography, and nano-imprint lithography [15,34]. Generally, in lithography techniques the pattern is transferred from a master to a replica. In photolithography, a light sensitive material (photoresist) is spin-coated on the substrate (for e.g., a silicon wafer). A photomask with a desired pattern is placed on the coated surface and exposed to UV light. With the use of a negative photoresist (for e.g., SU8), the regions exposed to the UV light form free radicals which cause cross-linking (polymerization) upon heating. Using a developer solution, the unexposed regions are washed away selectively, leaving the desired nano/micro scale structures on the substrate. Similarly, a positive photoresist (for e.g., diazonaphthoquinone molecules blended into novolac resin) can be used although the patterns formed in the process are reversed; regions of the positive photoresist exposed to light weaken and can be washed off with a solvent. Using photolithography, the shapes and sizes of the fabricated patterns are controllable. Furthermore, the fabricated template can be used as a mold and is reusable [133], thereby making the process cost-efficient. However, having a clean and flat starting surface is essential [134] which may require the need of a cleanroom.

Lithography techniques can be combined with each other or other techniques to provide SH surfaces. Ji et al. [135] proposed a hybrid technique for fabricating anti-corrosion epoxy thermoset (ET) coatings on cold rod steel electrodes. They used photolithography to fabricate micro-scale cylinders on a silicon wafer and colloid lithography to create nanoscale tips on the structures. The structures were then transferred to the ET coating using PDMS as the soft template (Figure 21c). The coating provided a WCA of 160° and stable anti-corrosion properties. In another study [136], a 50 μm gold layer was coated on the silicon wafer via electron beam evaporation to act as the conductive seed layer for the electroplating of nickel. Using photolithography, an array of holes was formed. The surface was then placed in the electroplating bath containing nickel sulfamate, boric acid, and a fluoride surfactant at constant pH and temperature. With the current density of 0.1 and 0.5 A/dm^2^, nickel mushroom microstructures were overgrown from the patterned holes. Acetone and oxygen plasma were used to strip the photoresist layer. PDMS was used as a soft mold in UV-nanoimprint lithography to replicate the mushroom structures. The samples were then coated with carbon through hot filament chemical vapor deposition, making the surface anti-bacterial against *E. coli* and *S. aureus*.

Durret et al. [137] demonstrated using nanoimprint lithography (NIL) and plasma etching processes with CF_4_/Ar on fluorinated ethylene propylene (FEP), polymethyl methacrylate (PMMA) and polyethylene terephthalate (PET) polymer films to fabricate hierarchical structures on surfaces (Figure 22). The plasma treatment step improved the superhydrophobicity of the surfaces. With a relatively short imprinting process time and a very short plasma treatment time on FEP and PMMA, a WCA of 160 ± 2° and a CAH of 2 ± 2° were achieved.

Jung et al. [138] reported using photolithography and deep reactive ion etching (DRIE) processes to fabricate hierarchical structures on a silicon master surface. To prevent the silicon master surface from damage, ultraviolet (UV) imprinting was applied on the surface to generate a polymeric master. A nickel seed layer enabled electrochemical deposition. Pulse-reverse-current (PRC) electrochemical deposition and subsequent metal-to-metal electrochemical deposition was used to fabricate a SH metal surface with micro-hierarchical structures. The fabricated SH surface showed a higher WCA (159.35 ± 1.02°) and higher anti-icing properties compared to the bare metal surface, a random nanostructured surface, and an ordered nanostructured metal surface.

### 3.10. Thermal Techniques

Recently, researchers have developed thermal techniques to create micro/nanostructures on the surface of various substrates. Hot embossing (also known as hot pressing or lamination templating) is a simple and common thermal nanoimprint process. A thin polymer film is heated to a temperature between its softening and melting points. Then the desired pattern is printed on the surface by a pressure that is significantly less than the molding pressure. After cooling, the polymer substrate is removed from the mold and the inverse template structure is transferred to the surface of the polymer (Figure 23) [139,140]. Because this technique relies on gently melting the desired surface of the substrate onto the template, a polymer is often used as the substrate. It is also important to note that the material used to make the stencil typically has a much higher melting point than the polymer.

Toosi et al. [141] used SS templates produced by femtosecond laser ablation to create periodic micron/submicron structures onto three polymers—high-density polyethylene (HDPE), polylactic acid (PLA), and medical-grade polyvinyl chloride (PVC). The choice of these three materials was based on their potential application as SH biomaterials in biomedicine and tissue engineering. The polymers were heated to temperatures just above their melting point and structures were patterned at pressures between 3 and 12 MPa. The WCA on HDPE and PLA was found to be over 160° while the CAH was under 10°. They also found that a re-entrant structure with a fibrillar morphology was produced on the surface of HDPE by applying a directional force during template removal from the replicated surface at high temperatures. Fibril formation was due to micro adhesion (chain adsorption) of HDPE chains on the metallic SS surface. This bio-inspired morphology possesses characteristics of superoleophobicity.

Hot embossing has the advantage of scalability and is therefore well suited for mass production of pattern surfaces from an original template with high reproducibility. Rather than allowing the mold to cool before removing the polymer, pulling the substrate while the mold is hot can produce high aspect ratio nanostructures on an inherently hydrophobic polymer such as HDPE thereby rendering the surface to be SH. A disadvantage of hot embossing is the need for multiple cycles to produce large quantities and the need for multiple processing steps. Typically, thermoplastics and elastomers, are used in this technique.

### 3.11. Perspective on Fabrication Techniques

Generally, a SH surface with effective, uniform, and durable structures fabricated by a cost-and time efficient, environmentally friendly technique is desirable for large-scale production. This tangible goal has yet to be achieved. Because of the high capital costs, some methods including laser processing, CVD, plasma processes, and lithography are not economically viable for large-scale production. However, the hybrid and modified versions of coatings, chemical etching, ED, and sol–gel processes are promising and reliable techniques for the fabrication of durable large-scale SH surfaces. In chemical etching processes, weaker acid solutions, assisted by surface modification with non-fluorinated agents are preferred to make this method more sustainable. EPD as a variation over conventional ED, can be a promising technique due to its potential to fabricate durable SH surfaces through a single step process. Considering all wet chemical techniques, the sol–gel method is the most effective, cost-efficient technique with minimal risk to human health and the environment. In cases where uniformity of the SH surface is not a concern, the sol–gel technique appears to be the most suitable and applicable for a wide range of materials.

Table 1 summarizes the various merits and limitations of the most common techniques used to fabricate SH surfaces. In addition, the applicability of the techniques to various materials are listed. We note that the table is not exhaustive, and exceptions may be present.

## 4. Characterization of the Static and Dynamic Wetting Properties of Surfaces

Several parameters are used to characterize the wetting behaviors of surfaces, including static, receding and advancing contact angles, the sliding (roll-off) angle, and adhesion (pull-off) and friction (lateral adhesion) forces. Temporal effects can affect these measurements and therefore the contact time between a water drop and the surface is also a factor to consider. The wetting properties of surfaces can be tuned by the judicious choice of the material, modifying the surface chemistry and controlling the surface morphology (i.e., surface roughness and asperities which trap air). The characterization of the surface morphology is typically done using either a surface optical or mechanical profilometer, scanning electron microscopy (SEM) [142], and transmission electron microscopy (TEM) [143]. To achieve greater lateral and vertical resolution, atomic force microscopy (AFM) [144,145,146,147] is commonly used with the added advantage that samples do not need to be conductive. Force measurements, with high sensitivity (±0.1 nN) [148,149], can also be performed with the AFM.

As demonstrated in the previous section, several studies have focused on exploiting a wide range of micro- and nanofabrication and surface modification techniques to create SH surfaces. Precise characterization techniques are needed to effectively differentiate their wetting properties. Conventional wetting characterization techniques (such as optical imaging of the drop shape or using a contact angle goniometer) are known to be prone to high uncertainty, especially in the case of SH surfaces [150,151]. Early generations of contact angle goniometers relied on the ability of the user to accurately, reproducibly, and reliably locate the 3-phase contact line and extract the contact angle using an in-built protractor within an eyepiece. Therefore, the accuracy and reproducibility of measurements were significantly impacted by a user’s expertise in taking such measurements, with the accuracy of ±2° [152].

Improvements to the technique include the automated axisymmetric drop shape analysis-profile (ADSA-P)). The analysis creates the best theoretical profile of a drop to determine the CA, surface area, drop volume and three phase contact radius. The use of cameras and built-in image processing software allow for an improved accuracy of ±0.3° [152,153]. The inclusion of high-speed cameras has offered detailed views of drop impacts on surfaces and the accompanied interface deformation, from which researchers have been able to extract the dependence of several material properties and key parameters on the wetting process.

However, even with the improved hardware and software for analysis, measurements are still susceptible to error. For example, the misplacement by just one pixel of the baseline (which is typically identified by the user) can yield errors greater than 10° in WCA on SH surfaces [154]. Figure 24 illustrates how for WCAs higher than 150°, if one misses only one pixel from the correct baseline, the errors can dramatically increase [154].

Direct force measurements of the interaction between a liquid drop and a surface have been explored as a supporting and/or alternative method to characterize the wetting properties of surfaces. Depending on the sensitivity of the technique, force-based measurement instruments are able to measure interaction forces between surfaces at the micrometer down to the nanometer lengthscale [150]. Furthermore, two SH surfaces with almost same static WCA can have entirely different CAH [155,156,157], further justifying the need for a complementary or additional measurement technique with even greater accuracy. In this section, an overview of the optical characterization of wetting properties of surfaces is presented followed by an overview of recent efforts in using force-based measurements. The measurement sensitivity associated with the force-based techniques along with the sample (i.e., liquid drop) size is also summarized.

### 4.1. Optical-Based Characterization

The most common way to evaluate the superhydrophobicity or liquid repellency of surfaces is through optical-based characterization, using a contact angle goniometer [158,159,160,161,162,163,164,165,166,167,168]. Hydrophilic and hydrophobic surfaces are defined by having WCAs in the range of 10° < *θ* < 90° and 90° < *θ* < 150°, respectively, and are the most commonly encountered surfaces in our surroundings [169]. Superhydrophilic and superhydrophobic surfaces have attracted more attention due to their potential applications in engineering and are characterized by WCAs in the range of 0° < *θ* < 10° and 150° < *θ* <180°, respectively [170,171,172]. In the case of SH surfaces, the water shedding or sliding angle (SA) is also an important parameter and typically, low sliding angles are desired for superior self-cleaning properties [155]. However, SA measurements can be influenced by the drop volume, how a drop is deposited on a surface, how long it remains on the surface before the measurement (resting time), and surface defects or imperfections. In addition, in the case that the system exhibits stick-slip motion, the SA cannot be measured by tilting the sample, thereby limiting the characterization of SH surfaces [173,174].

T. Nongnual et al. [175] estimated the sliding angle, using a drop shape analyzer (DSA-30, Kruss Scientific, Hamburg, Germany) with a computational algorithm written in MATLAB (Figure 25). The image analyzer used the brightness of a grayscale image by averaging the intensity of all pixels. The sliding (roll-off) angle was measured at the time that the dark shadow disappeared, corresponding to the displacement of the three-phase points by 1 mm. A linear relationship was defined between the tilt angle of the surface (0–90°) and the acquisition time with an angular speed of the rotor as the slope. Thus, the sliding angle was calculated precisely with an accuracy of 0.2°. The tilting rate was decreased and optimized by 0.1° per frame to avoid the overestimation of the SA [175].

The capillary technique is another precise method [176,177], from which the contact angles can be obtained with an accuracy of ±0.1° [178]. Since the novel work proposed by Lucan, Washburn, and Rideal on the capillary rise technique in the 1920s [179,180,181], contact angles have been inferred by the Lucas-Washburn-Rideal Equation (LWRE). However, using this equation leads to the overestimation of CA due to uncertainties in the capillary radius and the presence of gas bubbles [182]. Through the capillary technique, CA can be measured directly using an optical microscope [182] or determined from the capillary force and liquid bridge geometry based on Delaunay’s analytical solution [178]. In a typical experiment, a solid plate is vertically immersed into a liquid at a slow rate (0.008–0.9 mm/min). The resulting curvature at the three-phase contact line is monitored simultaneously [152,182]. The restrictions of the technique are errors caused by evaporation of the liquid and the transparency requirement of the surfaces or coatings [178].

A comprehensive understanding of the drop impact of SH surfaces is essential, especially for anti-icing [183], dropwise-condensation [184], and self-cleaning properties [185]. Such phenomena are mainly characterized by two dimensionless numbers including the Weber number ($We=\frac{\rho D_{0}U_{0}{}^{2}}{\gamma }$) and Reynolds number ($Re=\frac{\rho U_{0}D_{0}}{\mu }$), where ρ, γ, and μ are the density, the surface tension, and the viscosity of the drop, respectively. $D_{0}$ is the initial drop diameter and $U_{0}$ is the drop impact velocity [186]. We is the ratio of kinetic energy on impact to the surface energy and Re is the ratio of the inertial force to the viscous force [187]. Depending on We number, different drop impact scenarios can occur. Changing the We number can result in deposition, rebounding, pinning/sticking, or splashing/fragmentation. In addition, the drop impact phenomenon is dependent on $Q=\frac{d_{jump}}{d_{max}}$ and the contact time, where $d_{jump}$ is the lateral diameter of the drop at the detachment from the surface and $d_{max}$ is the maximum lateral extension diameter of the drop.

In a typical experiment, from a predetermined height (thereby varying the impact velocity), a drop with a specific diameter is released from a syringe pump needle. The drop impact behavior is monitored by a high-speed camera (Figure 26) [188]. Post impact dynamics are not only a function of density, impact velocity and viscosity but also a function of surface roughness, geometry, wettability, and mechanical strength [188]. Y. Shen, et al. [189] achieved the minimum contact time on a complex textured SH surface. Generally, the contact time of an impact droplet includes spreading and retracting time. The spreading time is constant and under a certain kinetic condition (initial diameter $D_{0}=2\;\mathrm{mm}$ and initial impact velocity $V_{0}=0.1\;\frac{\mathrm{mm}}{s}$), it is equal to 5.5 ms. Hence, the only way to reduce the contact time is to shorten the retracting time. Through the fabrication of a complex surface with three-forked, cross-shaped, and five-forked microstructures, the retracting time was reduced to almost zero.

Table 2 summarizes some of the key experimental parameters and speed specifications of the high-speed cameras used in the studies described above. Drop sizes are typically in the range from sub-10 µL to ~10 µL. The small size of the drop minimizes the effect of gravity on the shape of the drop [190]. In other words, when the drop diameter is less than the capillary length ($l=(\frac{\gamma }{\rho g}{)}^{\frac{1}{2}})$), the drop is not distorted from its spherical shape [191]. Typical contact times are in the order of milliseconds and camera speeds ranged from 3000–20,000 fps [192,193,194,195].

### 4.2. Force-Based Characterization

When theories based on van der Waals and colloidal interactions were proposed to encompass material science, rheology, lubrication, cell interactions, phase transition, and DNA binding, it was essential to develop surface force measurements to test such theories [196]. High-resolution force measurements were performed by Tomlinson in 1928 [197]. The motivation for the work was to measure the adhesion force between fibers or spheres of glass and to establish the index of the decay law [196]. The invention of techniques such as atomic force microscopy, total internal reflection microscopy, and colloidal particle scattering allowed for sufficient force sensitivity to study interactions on microscopic surfaces [196].

Several techniques have been developed or adapted to measure the adhesion, snap-in, pull-off, and friction forces of liquid drops on surfaces such as the use of tribometers [198,199,200], the surface forces apparatus (SFA) [201,202], atomic force microscopy [148,203], the centrifugal adhesion balance (CAB) [204,205,206], and microbalances including micro-electromechanical, micro-electrochemical, and microelectronics [107,207,208,209,210,211,212,213,214,215]. A Taylor–Couette cell has also been used to measure drag [216]. Laser defection systems [217,218] have been used for friction measurements and a force tensiometer [219] has been used to measure adhesion forces and surface tension. In the next sections, such force-based instruments and accompanying techniques are discussed.

#### Sensor-Based Characterization

Sensor-based instruments are defined by an in-built spring/cantilever whereby its deflection is converted to the force and digitally recorded. In our previous work [174], the sliding angles of water drops on surfaces were determined from friction force measured by a nanotribometer (Figure 27). The sensor allowed for force measurements in the range of ±10 mN with ±1 µN sensitivity, enabling the user to apply and record various lateral (i.e., friction) and vertical (i.e., adhesion) forces. In a typical measurement, a water drop of predetermined volume (20 μL) is placed on a copper ring drop holder with an inside diameter of 1.7 mm. The drop is then contacted with the surface, a preload is applied, and the drop is allowed to equilibrate for 20 s. The probe is then moved at a constant speed of 0.1 mm/s while maintaining the applied load. During the sliding step, the normal and friction forces are measured until the drop is finally retracted from the surface (Figure 27C). This technique successfully predicts the sliding angle of water drops on surfaces, using the equation: $f_{\left|\right|}=mg\sin\left(\alpha \right)$ (Figure 27A), where, $f_{\left|\right|}$ is the kinetic friction force measured, m is the mass of the drop, g is the gravitational constant and α is the sliding angle [174]. The probe that holds the water drop can also be a capillary tube, as used by Qiao et al. [198], to measure friction forces resulting from a sliding water drop in a linear reciprocating mode on a microstructured solid surface. This system was able to record the lateral resistance force due to the liquid–solid interface and lateral displacement of the capillary tube simultaneously. As shown in Figure 28, a capillary glass tube was used to hold and slide a 2 µL drop on the surface. The drop was dispensed on the end of the tube using a micropipette. The setup included two independent dual-beam cantilever sensors to obtain friction and normal forces simultaneously.

Microbalances are also sensor-based instruments capable of measuring adhesion (pull-off) forces to characterize the liquid repellency of surfaces. Samuel et al. [212] investigated the wetting properties of 20 different polymers and SH surfaces. They used an electromechanical system to measure snap-in and pull-off (adhesion) forces. In a typical experiment, a water drop was placed on a metal ring connected to a cantilever (Figure 29). The stage holding the surface was then moved upward at a speed of 0.03 mm/s until contact was made with the water drop. The resulting force measured was referred to as the “snap-in” force. At this point, the surface was again moved upwards with the speed of 0.01 mm/s over 0.1 mm. The surface was then retracted at a speed of 0.01 mm/s. The force measured when the drop detached was referred to as the “pull-off” force or “adhesion force”. It was found that the adhesion force was not correlated to WCAs. Interestingly, there was a correlation between the pull-off force and the receding contact angle; as the receding contact angle increased, the pull-off force decreased.

### 4.3. Hybrid Characterization

Researchers have also explored hybrid characterization systems in which optical techniques are used to measure/infer adhesion and friction forces between a water drop and a surface, for e.g., cantilever sensors, centrifugal adhesion balance (CAB) and capillary sensors. The most common hybrid-based surface characterization method is the cantilever sensor technology, including atomic force microscopy (AFM) [220], mass spectroscopy, and magnetic resonance force microscopy. In order to compare the test results of various research groups, reliable calibration is required [221].

Microfabricated cantilevers are mostly used to measure the topography of surfaces such as those used in scanning force microscopy (SFM) or atomic force microscopy (AFM) [222,223]. AFM has been developed into one of the most sensitive and adaptable surface-characterization apparatus to explore surfaces at the molecular and atomic level. Cantilevers are typically silicon-based, rectangular-shaped and less than 1 µm thick. The bending of the cantilever, as a result of interactions forces with a surface, is detected by the deflection of a laser beam. The advantages of such miniaturized sensors include their small size, fast response time, and high sensitivity [224]. The adhesion and friction forces between the surface and the liquid interface can be measured with the sensitivity of 0.1 nN by attaching a liquid drop on a tipless AFM probe [148,149]. The AFM technique may not be suitable for the characterization of wetting properties of all surfaces (for e.g., structured surface with feature dimensions larger than or comparable to the liquid drop size), however, its sensitivity is considerably higher compared to all other methods.

R. Tadmor and his group [204] invented the CAB to measure lateral adhesion forces of liquid drops and investigated the effects of load and resting time on the friction force. Such experiments cannot be run on conventional tilt stages since loads and lateral forces are coupled and the range of an applied lateral force is limited between 0 and the weight of the drop (corresponding to a tilt angle of 0° and 90°, respectively). They proposed that a higher normal load can lead to lateral force reduction even though the contact area increases. Additionally, as the resting time between a hexadecane drop and the octadecyl trimethylammonium treated mica surface increased, friction forces increased due to time-dependent molecular reorientation and hydrophobic recovery of the surface [225].

In another study [205] they used a customized CAB to predict sliding angles of water drops on surfaces accurately (Figure 30 [205,206]). The instrument has a centrifugal arm rotating in the horizontal plane. At the end of the centrifugal arm, a chamber is attached in which a camera is mounted to monitor the drop’s motion. As the shaft rotates, the lateral adhesion force increases due to the increasing acceleration. Once the drop detaches from the surface, the acceleration is recorded and used to calculate the SA [205]. It is worth mentioning that this measured lateral adhesion force corresponds to the static friction force.

The capillary force sensor technique which includes the micropipette force sensor (MFS) [226,227,228] and the droplet force apparatus (DFA) [229] is another method for force measurements which can provide the sliding forces of a water drop with high sensitivity. The static force, SF, is defined as the force applied to a stationary drop until it overcomes the threshold force, *F_THRD_*, at which point the drop starts sliding. A lower kinetic force, KF, is sufficient to maintain the sliding of the drop [217]. Such forces can be differentiated with high resolution because of the high sensitivity of capillary sensors. In a typical capillary sensor force measurement, a solid substrate is moved by a linear stage driven by a step motor as shown in Figure 31a,b [227]. A camera is used to record and measure the deflection of the capillary tube. The basic principle behind these techniques [230,231,232] is the use of a capillary tube to connect a liquid drop to a spring (or the capillary tube itself can serve as the spring) with a known spring constant, *k*. The force exerted on the drop can be calculated using Hooke’s law, i.e., F=kΔx where Δx is the deflection of the capillary tube or spring (Figure 31b) [227].

Kui Shi et al. [173] utilized a capillary sensor to measure the friction force between a water drop and solid surface. By recording the position of the capillary tube while the drop is sliding, the capillary deflection was converted to the friction force. Generally, when the droplet does not slide on a surface at tilt angle of 90°, meaning its weight is too small to overcome the threshold force, SA measurements with conventional methods cannot differentiate the wetting properties of surfaces. Capillary sensors not only differentiate the hydrophobicity of such surfaces but also provide static, transition, and kinetic regimes with the sensitivity of 0.7–2 μN.

### 4.4. Perspectives on Surface Wetting Characterization Techniques

Table 3 summarizes the techniques, corresponding sensitivities, and typical drop volumes used in the references discussed in this section. Drop sizes are typically less than 10 µL in such measurements (comparable to drop sizes used in optical characterization techniques) which minimizes the effect of gravity on the spherical shape of the drop.

Optical-based techniques have the advantages of being convenient, fast, and cost-efficient. However, because of their uncertainties in the characterization of SH surfaces, force-based techniques are recommended. The instruments measuring adhesion forces including microelectromechanical balances and micropipette force sensors have the advantage of providing high sensitivity, whereas the dependency of the drop shape on ring size and drop volume can cause irreproducibility [186]. The centrifugal adhesion balance is an accurate characterization method being able to measure static friction force with the sensitivity of ±1 μN. It is worth noting that surfaces may contain imperfections, resulting in nonhomogeneous wetting properties. As a result, measurement techniques which rely on collecting data under static conditions or at the onset of motion are only considering the wetting properties of drop at that particular location. Due to the stochastic nature of coatings on SH surfaces [228], measuring the dynamic friction force is recommended to average out the influence of imperfections on surfaces. Capillary force sensors and tribometers have the advantage of resolving the various stages in friction processes (static, kinetic and stick-slip behaviors), snap-in and adhesion forces of a water drop. Such sensors measure the aforementioned forces directly with sufficient sensitivity (0.7–2 μN) making them, in our opinion, the most promising techniques among all characterization methods.

## 5. Conclusions

The field of biomimetics is increasingly gaining popularity as scientists continually discover how nature has evolved organisms to overcome challenges. However, challenges still exist in mimicking the desired properties. In the case of the lotus leaf, numerous fabrication techniques have been proposed to create SH surfaces on metals, polymers, fabrics, glasses, and ceramics. As researchers continue to explore novel fabrication techniques to create SH surfaces, special consideration should be given to: (i) the scalability of the technique; (ii) the durability of the resulting SH surfaces; (iii) the potential environmental impact of the chemicals used in the process; and (iv) the cost of the instruments and materials used in the process to make the technique commercially viable. As for the characterization of the wetting properties of SH surfaces, optical-based techniques will likely remain relevant in future studies because of the simplicity, availability, and ease of operation of the technique. However, force-based techniques provide higher sensitivities and may be required to truly differentiate the wetting properties of SH surfaces with similar WCAs, SAs, and CAH. Although the fundamental science behind superhydrophobicity and the roles that surface chemistry and surface morphology play in tuning superhydrophobic properties are well understood, further research is still needed to fully understand their origins and their relative contributions to CAH.

## Figures and Tables

**Figure 1 biomimetics-07-00196-f001:**
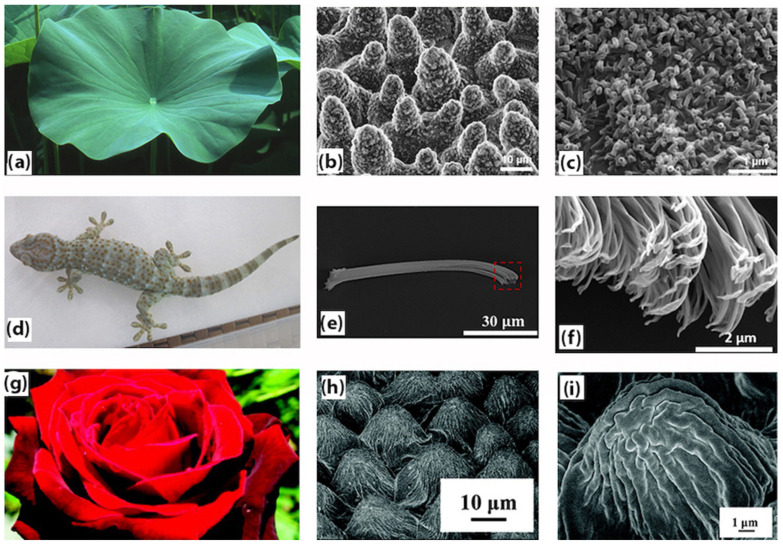
Images of various plants and animals showing their microscale and nanoscale structures: (**a**) image of a Lotus leaf; (**b**) scanning electron microscopy (SEM) image of the upper side of a Lotus leaf prepared by ‘glycerol substitution’ shows the hierarchical surface structure consisting of papillae, wax clusters and wax tubules; (**c**) wax tubules on the upper leaf side [6]; (**d**) image of a Tokay Gecko; (**e**) SEM image of a single seta; (**f**) SEM image of clusters of spatulae branching out the tip of a single seta (Reprinted from [7] with permission from Elsevier); (**g**) image of a red rose; and (**h**,**i**) SEM images of the surface of a red rose petal, showing a periodic array of micropapillae and nanofolds on each papillae top (Reprinted with permission from [8]. Copyright {2008} American Chemical Society).

**Figure 2 biomimetics-07-00196-f002:**
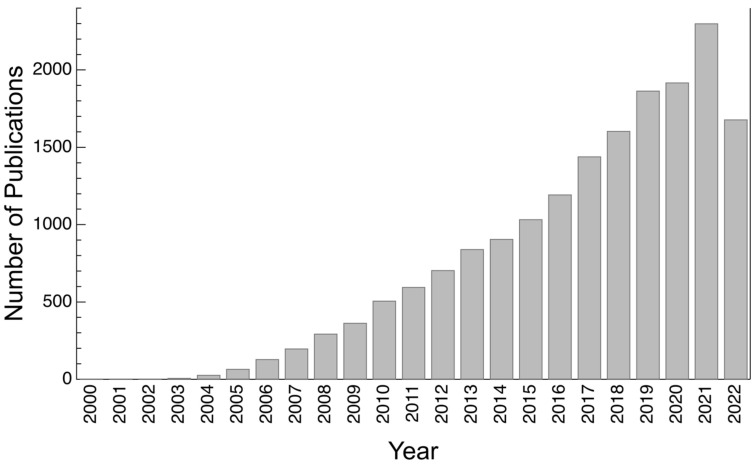
Plot showing the rising number of publications indicative of an increasing interest in the area of superhydrophobicity. A keyword search of “superhydrophobic” was performed using the Web of Science search engine. Meeting abstracts and proceedings papers were excluded from the search.

**Figure 3 biomimetics-07-00196-f003:**
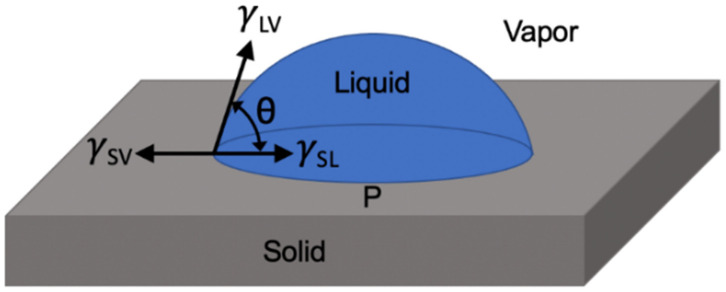
Schematic illustration of a liquid drop on a smooth solid surface surrounded by a vapor phase. The interfacial energies associated with the 3 phases include solid–liquid, $\gamma_{SL}$, solid–vapor, $\;\gamma_{SV}$, and liquid–vapor, $\gamma_{LV}$. The WCA, *θ*, is measured at the 3-phase contact line, P, from the solid–liquid interface to the liquid–vapor interface within the liquid phase.

**Figure 4 biomimetics-07-00196-f004:**
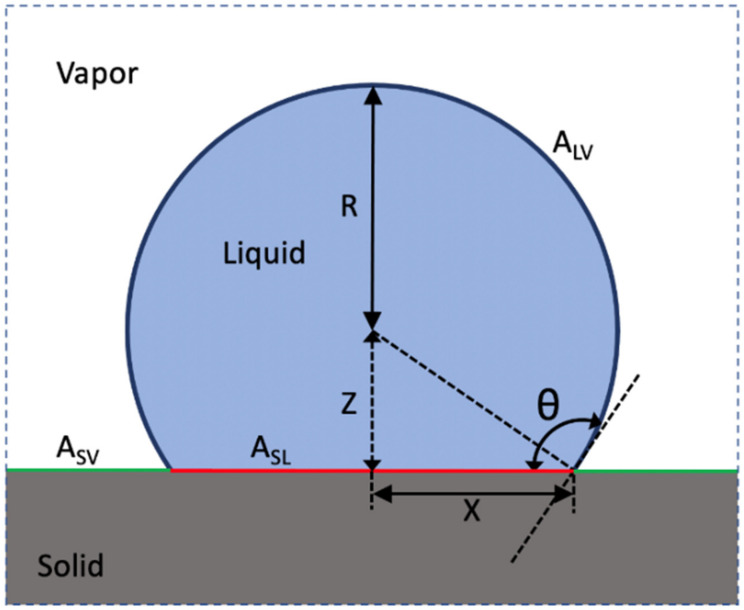
Schematic illustration of a liquid drop on a smooth solid surface surrounded by a vapor phase. The dashed box denotes the system. The liquid drop can either wet or de-wet the solid surface (i.e., a change in *θ* and *R*) with the constraint of a fixed liquid drop volume. Equilibrium is achieved when the free energy of the system is minimized.

**Figure 5 biomimetics-07-00196-f005:**
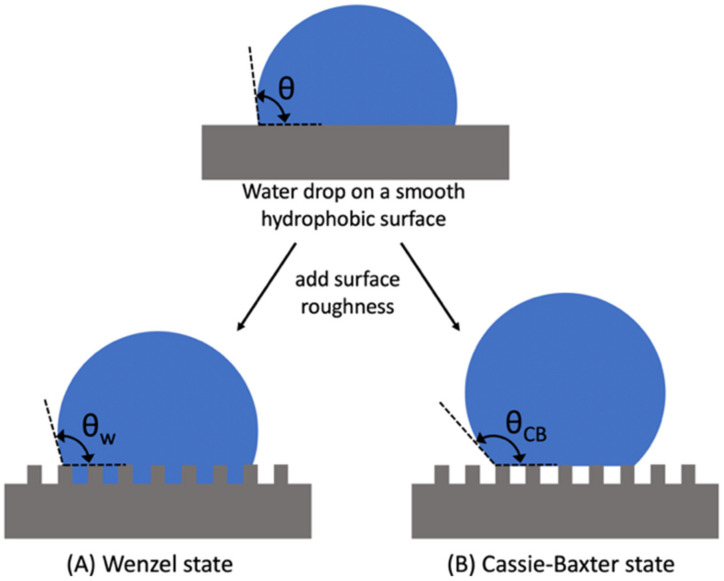
Schematic illustration of how the addition of roughness to a hydrophobic smooth surface can lead to an increase in the static contact angle as predicted by the (**A**) Wenzel; and (**B**) Cassie–Baxter model.

**Figure 6 biomimetics-07-00196-f006:**
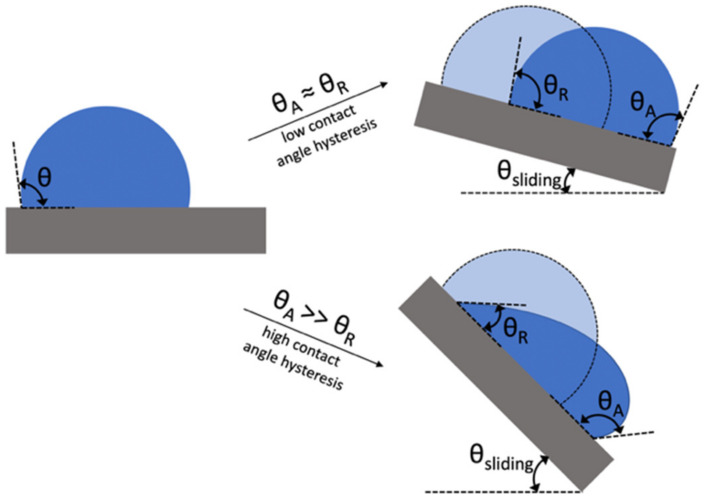
Schematic illustration of the dynamic (i.e., advancing and receding) contact angles resulting from the motion of a water drop on a tilted surface. A larger contact angle hysteresis (i.e., difference between $\theta_{A}$ and $\theta_{R}$ leads to requiring a larger tilt to initiate drop sliding.

**Figure 7 biomimetics-07-00196-f007:**
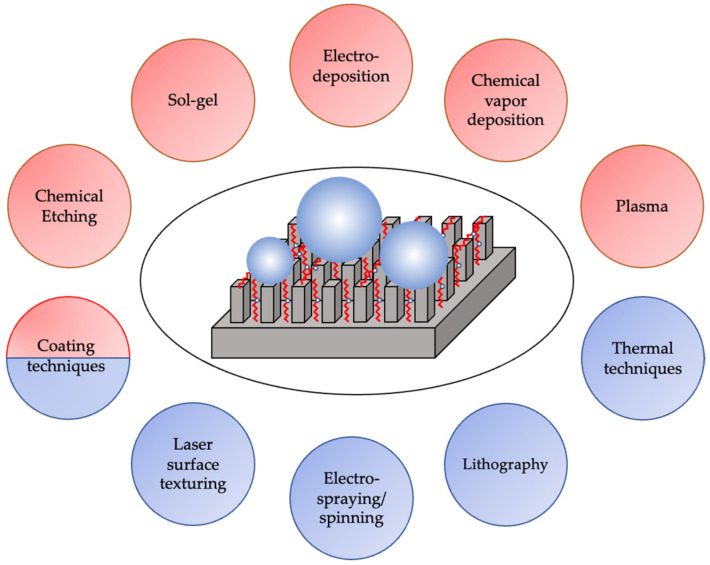
An overview of fabrication techniques of SH surfaces, categorized into physical (blue-color) and chemical techniques (red-color).

**Figure 8 biomimetics-07-00196-f008:**
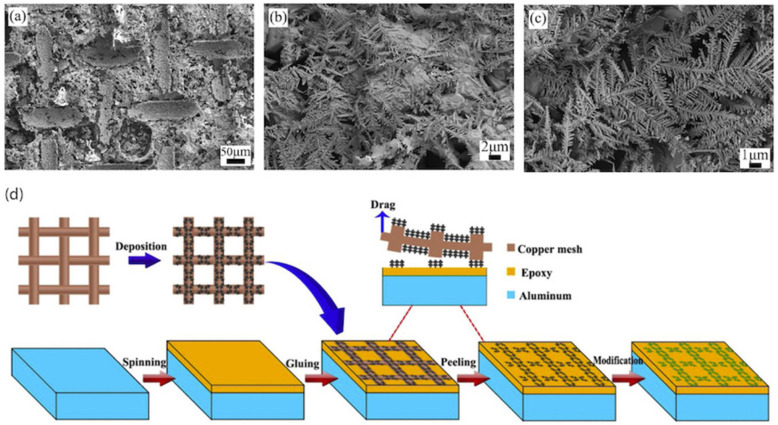
(**a**–**c**) SEM images of a modified aluminum surface at various magnifications; and (**d**) schematic illustration of the process involving spin-coating epoxy resin on aluminum (Reprinted from [35] with permission from Elsevier).

**Figure 9 biomimetics-07-00196-f009:**
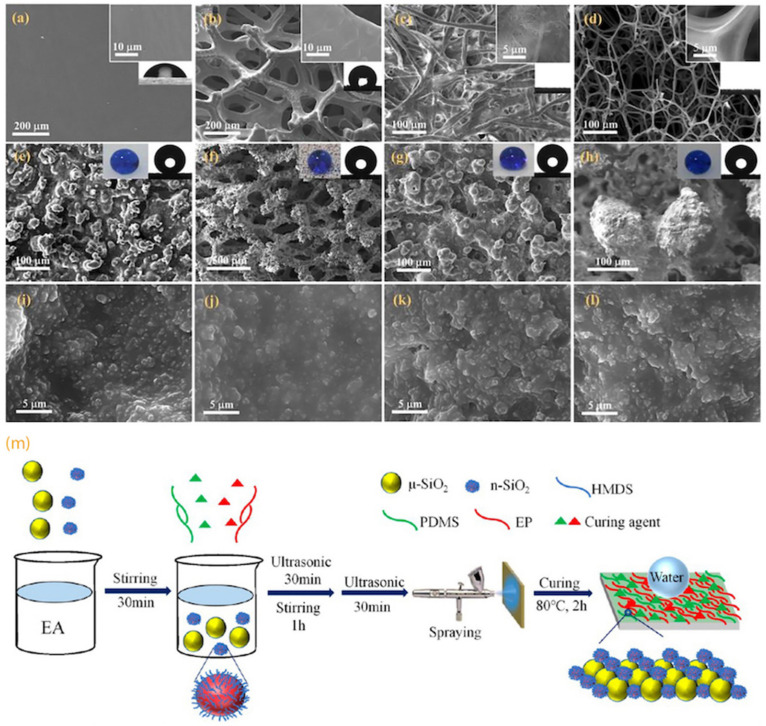
SEM images of untreated of: (**a**) glass; (**b**) copper foam; (**c**) filter paper; (**d**) sponge; (**e**–**h**) surface morphology of the corresponding epoxy resin+PDMS@SiO_2_-coated samples; and (**i**–**l**) SEM images of the coated samples at higher magnification (Reprinted from [44] with permission from Elsevier). (**m**) Schematic illustration of the spraying process (Reprinted with permission from [43]. Copyright {2021} American Chemical Society).

**Figure 10 biomimetics-07-00196-f010:**
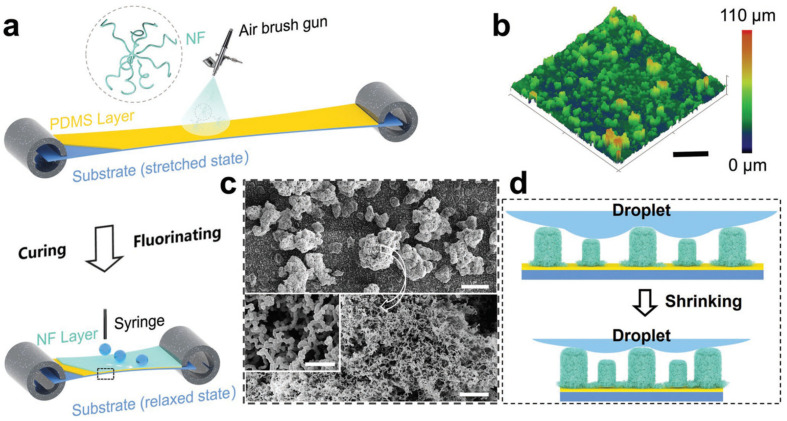
(**a**) Schematic illustration of the fabrication process of the nanofilament-structured and stretchable superamphiphobic (NFSS) surface; (**b**) confocal 3D images of the NFSS surface. Scale bar = 200 μm; (**c**) SEM images of the NFSS surface at various magnifications. Scale bar = 50 μm (**top**), 2 μm (**bottom**), and 500 nm (inset); and (**d**) schematic illustration of microstructures on the NFSS after releasing the surface from the stretched state (Reproduced from [48] with permission from John Wiley and Sons).

**Figure 11 biomimetics-07-00196-f011:**
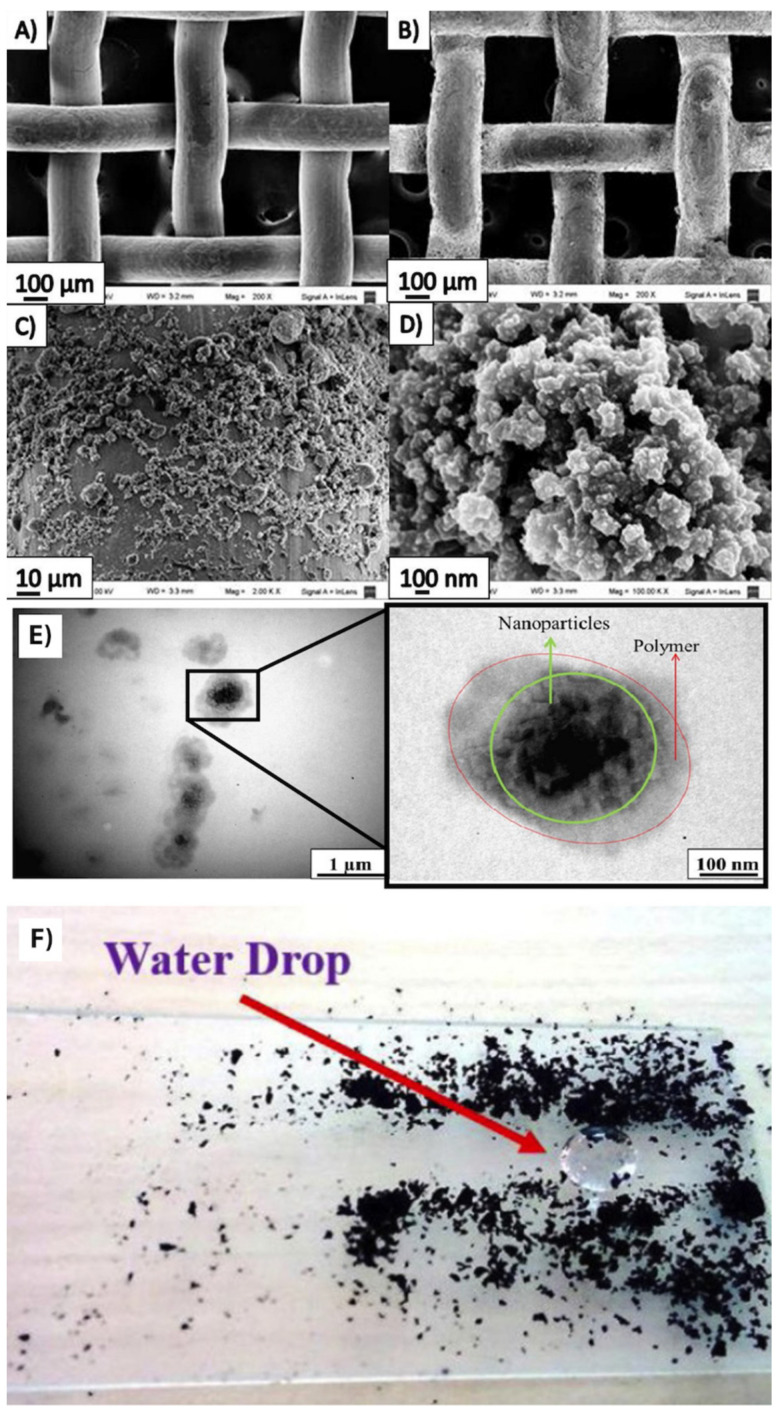
(**A**,**C**) Pristine brass mesh; (**B**,**D**) brass mesh coated with SiO_2_-PDMS-PVDF under different magnifications; (**E**) TEM images of the SiO_2_-PDMS-PVDF composite and a polymer coating around the SiO_2_ nanoparticles (Reprinted from [32] with permission from Elsevier); and (**F**) self-cleaning property of a SH glass surface modified with silica nanoparticles (Reprinted from [50] with permission from Elsevier).

**Figure 12 biomimetics-07-00196-f012:**
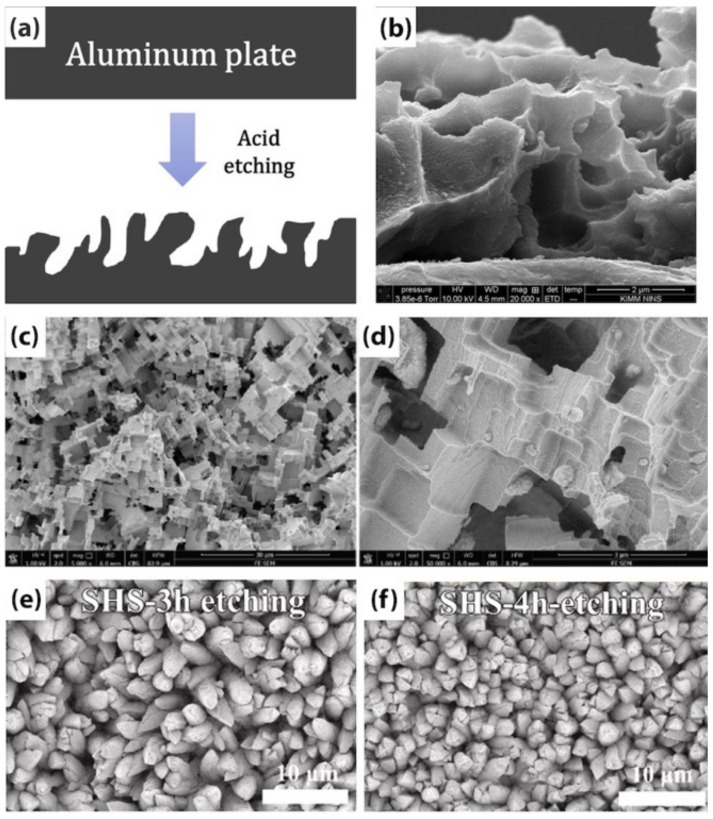
(**a**) Schematic illustration of creating surface roughness by chemical etching of an Al plate with an acidic solution of HCl, isopropanol and DI water; (**b**–**f**) the SEM images of hierarchical structures on an Al surface after modification in (**b**) HCl solution (Reprinted from [67] with permission from Elsevier); (**c**,**d**) FeCl_3_ solution (Reprinted with permission from [66]. Copyright {2021} American Chemical Society); and (**e**,**f**) ammonia solution (Reprinted from [68] with permission from Elsevier).

**Figure 13 biomimetics-07-00196-f013:**
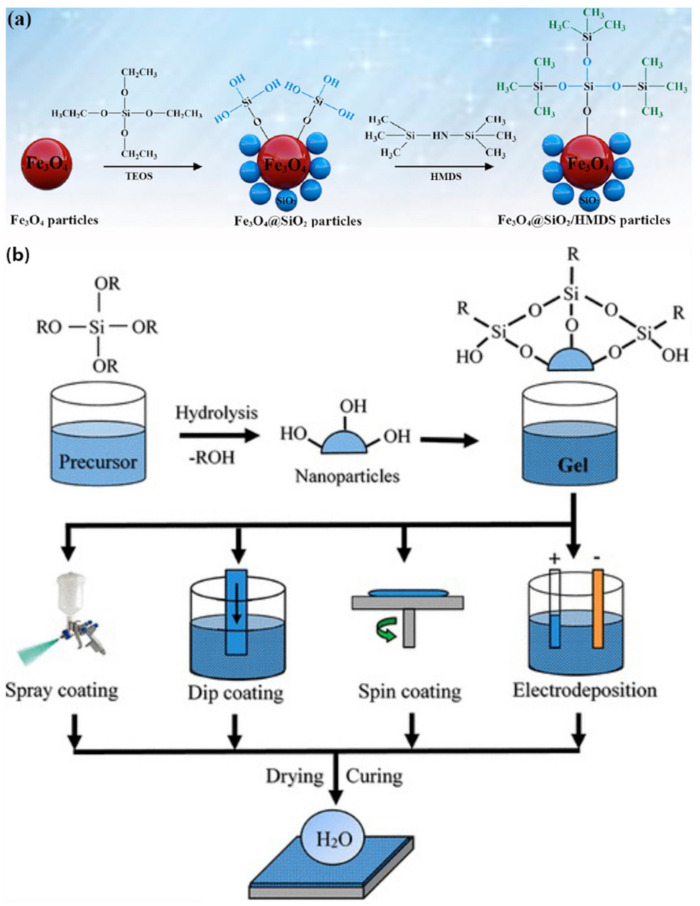
(**a**) Schematic of the one-pot sol–gel fabrication method to obtain Fe_3_O_4_@SiO_2_/HMDS particles (Reprinted with permission from [80]. Copyright {2020} American Chemical Society); and (**b**) illustration of sol–gel process followed by other fabrication methods for coating [84].

**Figure 14 biomimetics-07-00196-f014:**
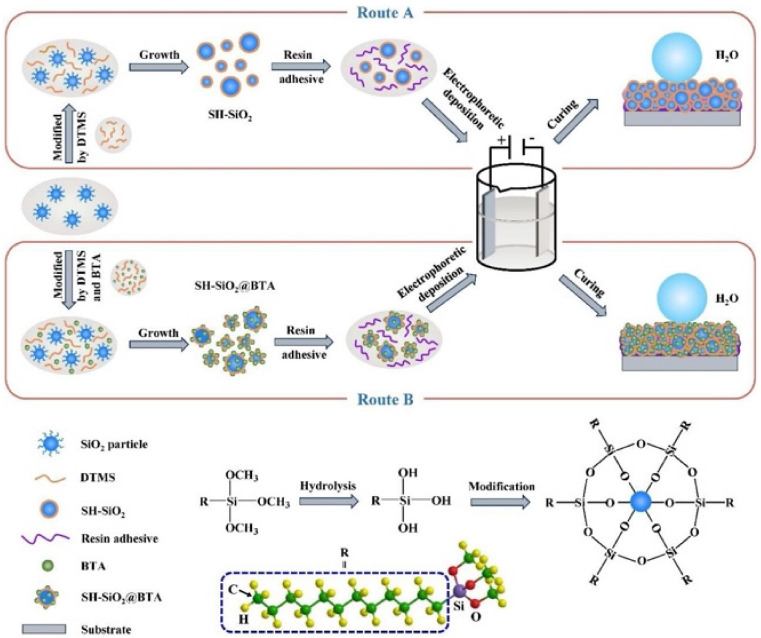
Schematic illustration of the EPD process for the fabrication of SH-SiO_2_ (Route A) and SH-SiO_2_@BTA (Route B) nanoparticle-based SH surfaces (Reprinted from [98] with permission from Elsevier).

**Figure 15 biomimetics-07-00196-f015:**
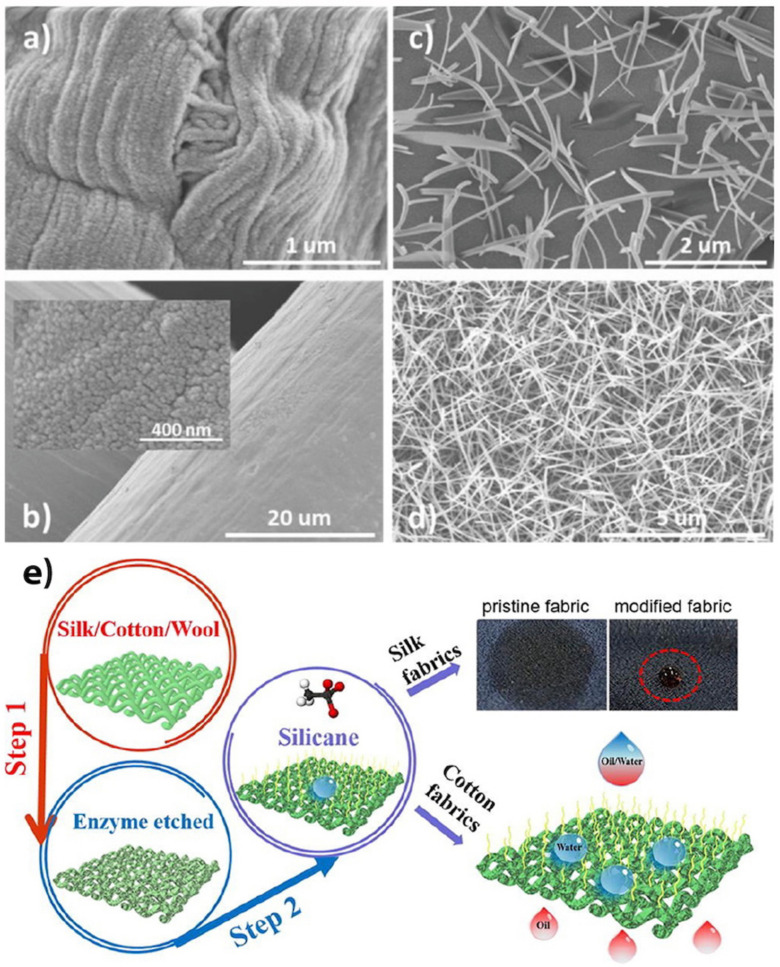
SEM images of (**a**) cellulose filter, and (**b**) SS membrane after the first PECVD to form TiO_2_ nanofibers as the nucleation layer (step 1); (**c**,**d**) Top view SEM images of H_2_Pc nanowires created on TiO_2_ nucleation layer under the growth rate of <0.5 Ås^−1^ and >1.5 Ås^−1^, respectively. (Reproduced from [102] with permission from John Wiley and Sons). (**e**) The enzyme etched fabric is silanized via CVD to reduce the surface energy for oil/water separation application. (Reprinted from [105] with permission from Elsevier).

**Figure 16 biomimetics-07-00196-f016:**
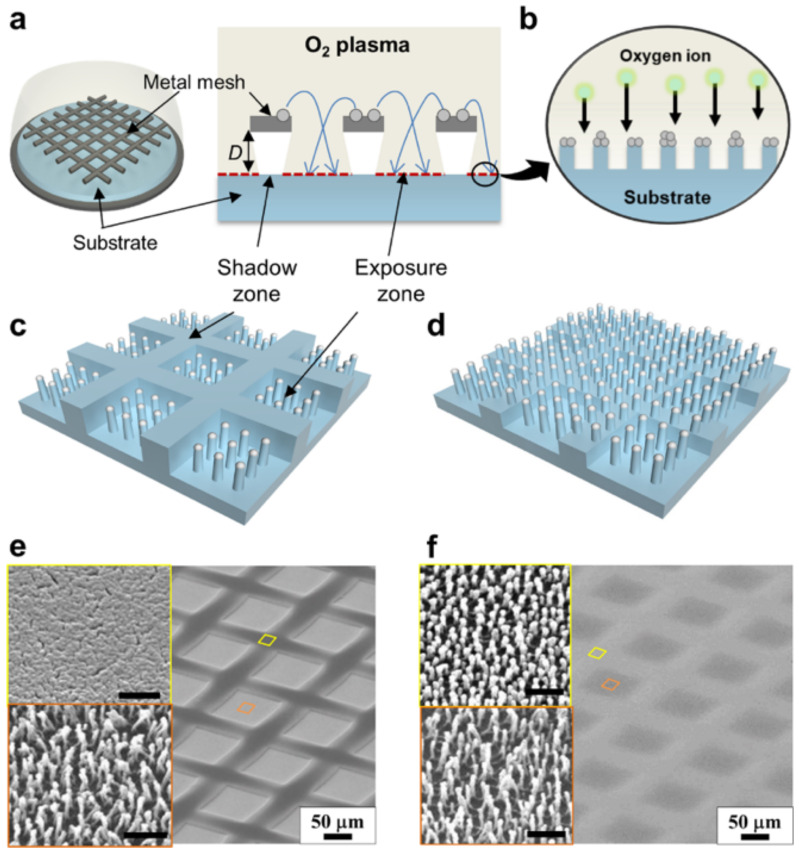
Schematic illustration of hybrid patterning process using dual scale metallic mesh; (**a**) shadow zone with microscale structures, (**b**) nanoscale etching mask, (**c**,**e**) hybrid textures including micro walls and nanostructures, and (**d**,**f**) hierarchical pillars consisting of low micro walls and nanostructures with shorter gap distances. Scale bars in insets are 500 nm [107].

**Figure 17 biomimetics-07-00196-f017:**
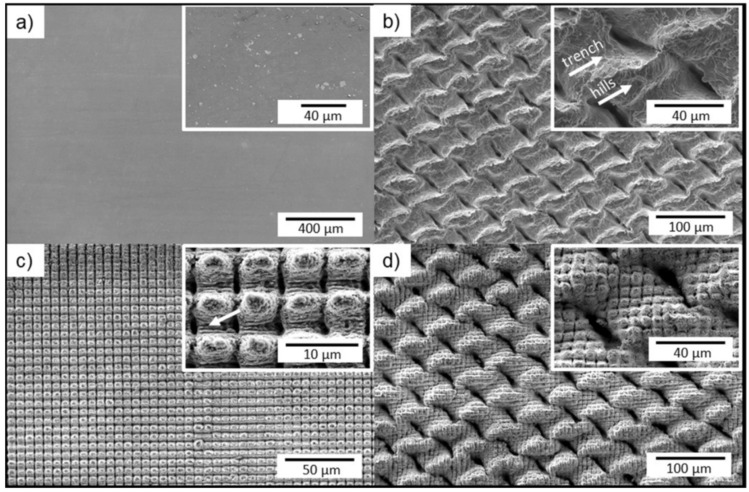
SEM images of (**a**) An original Al 2024 reference, (**b**) The mesh-like structure processed with DLW, (**c**) The pillar-like DLIP structures with a spatial period of 7 μm (the arrow in the inset indicates the line-like LIPSS), and (**d**) A hierarchical structure resulting from the combination of DLW and DLIP (Reprinted from [117] with permission from Elsevier).

**Figure 18 biomimetics-07-00196-f018:**
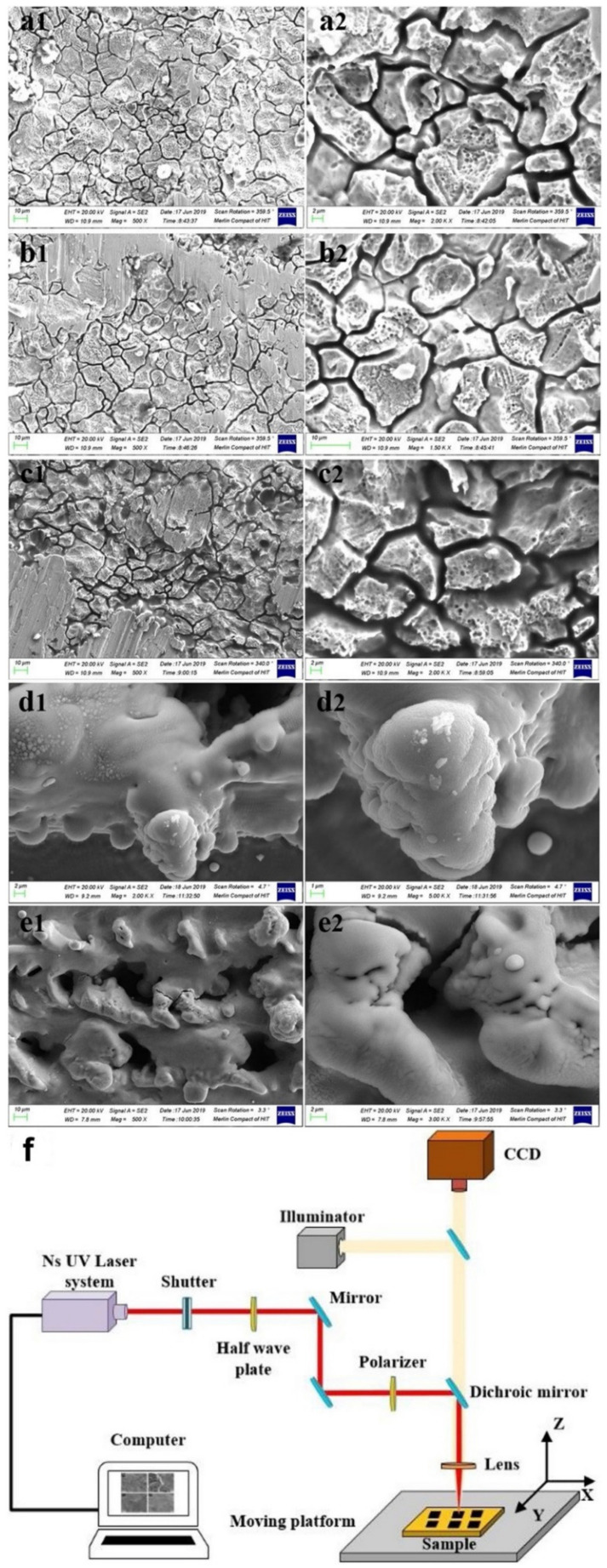
SEM images of laser textured surface at scanning speed of 20 mm/s with different fluence (**a**) 2.69 J/cm^2^ (**b**) 3.96 J/cm^2^ (**c**) 6.28 J/cm^2^ (**d**) 8.14 J/cm^2^ (**e**) 9.55 J/cm^2^, (**a2**–**e2**) are magnified SEM images, respectively. (**f**) Schematic illustration of a nanosecond UV laser texturing system (Reprinted from [123] with permission from Elsevier).

**Figure 19 biomimetics-07-00196-f019:**
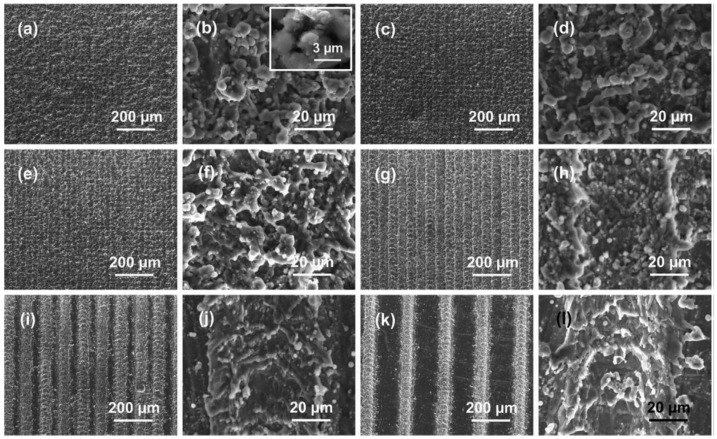
SEM images of Al surfaces fabricated by the nanosecond laser with different scan spacing. (**a**,**b**) 10 mm. (**c**,**d**) 20 mm. (**e**,**f**) 30 mm. (**g**,**h**) 50 mm. (**i**,**j**) 100 mm. (**k**,**l**) 200 mm. (Reprinted from [31] with permission from Elsevier).

**Figure 20 biomimetics-07-00196-f020:**
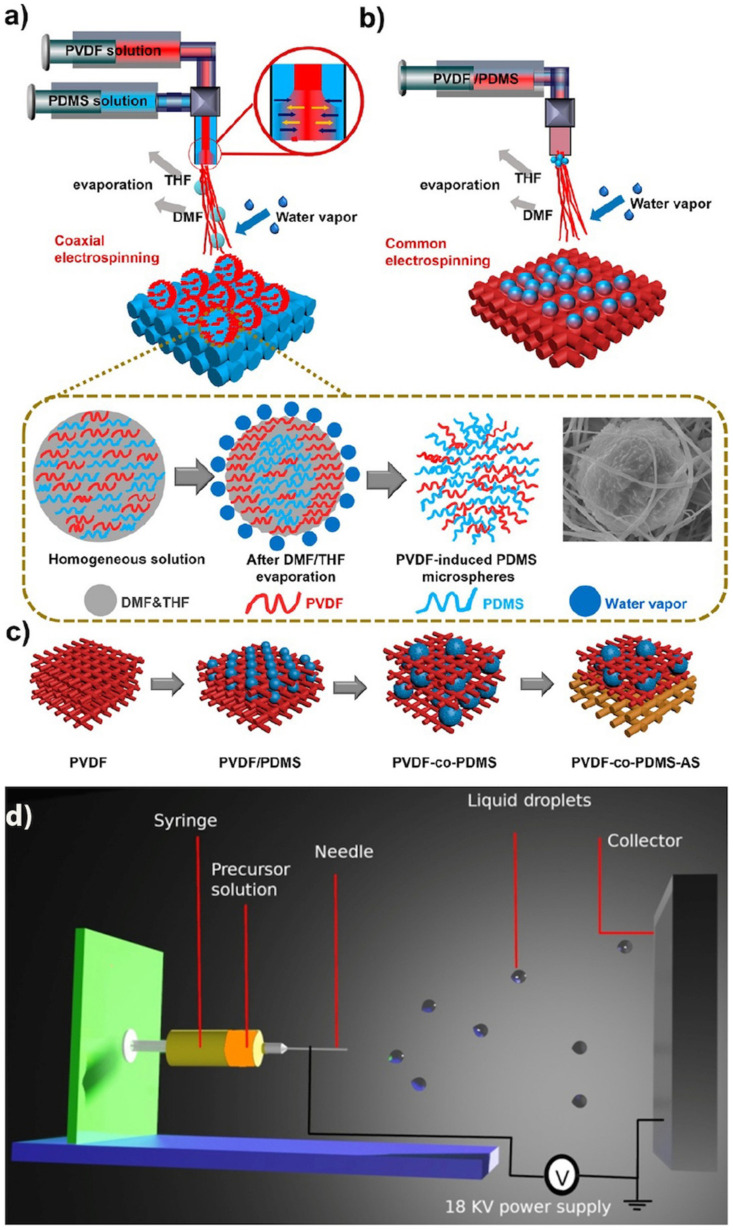
Schematic representation of (**a**) the coaxial and (**b**) conventional electrospinning process for the fabrication of nanofiber membrane, (**c**) fabrication steps of the asymmetric PVDF-co-PDMS nanofiber membranes (PVDF-co-PDMS-AS). The red nanofibers, blue microspheres, and brown substrate represent the PVDF selective layer, PDMS-based nanoparticles, and PVDF substrate, respectively. (Reprinted with permission from [129]. Copyright {2021} American Chemical Society). (**d**) A schematic of an electrospraying process. (Reproduced from [130] with permission from John Wiley and Sons).

**Figure 21 biomimetics-07-00196-f021:**
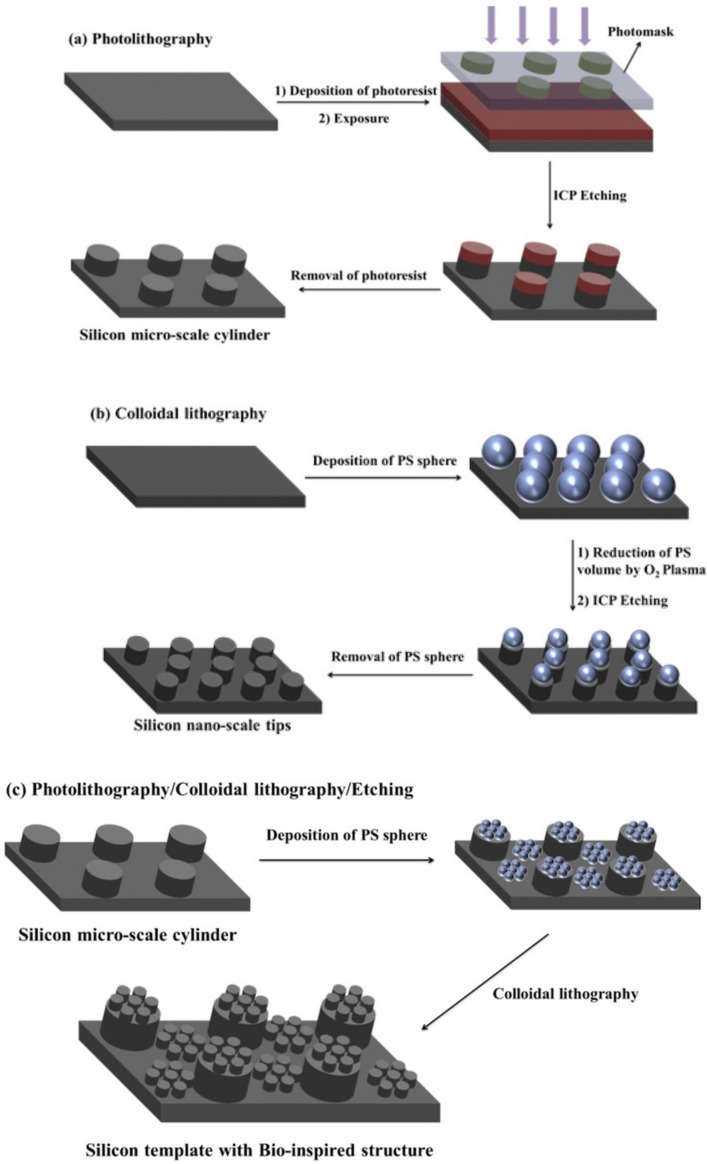
Schematic illustration of the fabrication of micro-, nano-, bio-inspired structures on a Si wafer hard template (Reprinted from [135] with permission from Elsevier).

**Figure 22 biomimetics-07-00196-f022:**
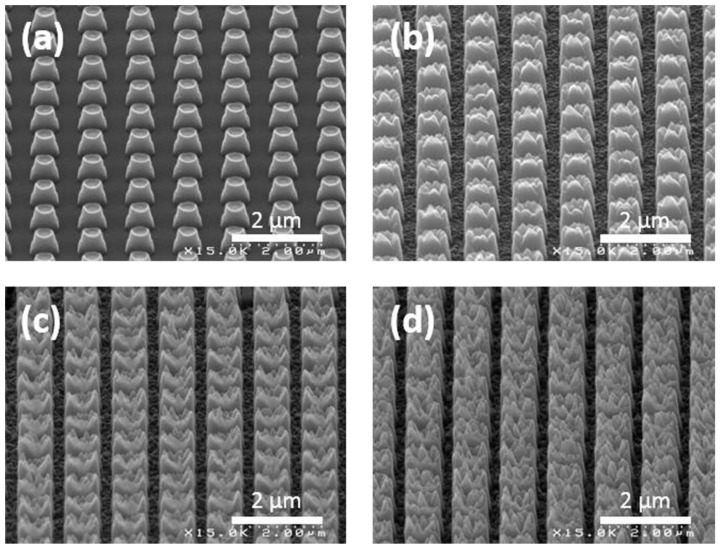
SEM images of (**a**) a native nanoimprinted film of FEP. Roughened dots were obtained by plasma treatment at 1000 W (**b**), 1600 W (**c**) and 1800 W (**d**). (Reprinted from [137] with permission from Elsevier).

**Figure 23 biomimetics-07-00196-f023:**
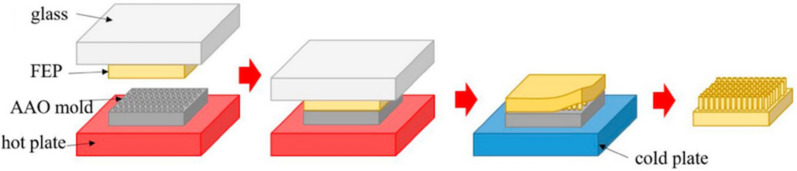
Schematic illustration of hot embossing technique [140]. A fluorinated ethylene propylene (FEP) sheet is pressed on a heated anodic aluminum oxide (AAO) membrane.

**Figure 24 biomimetics-07-00196-f024:**
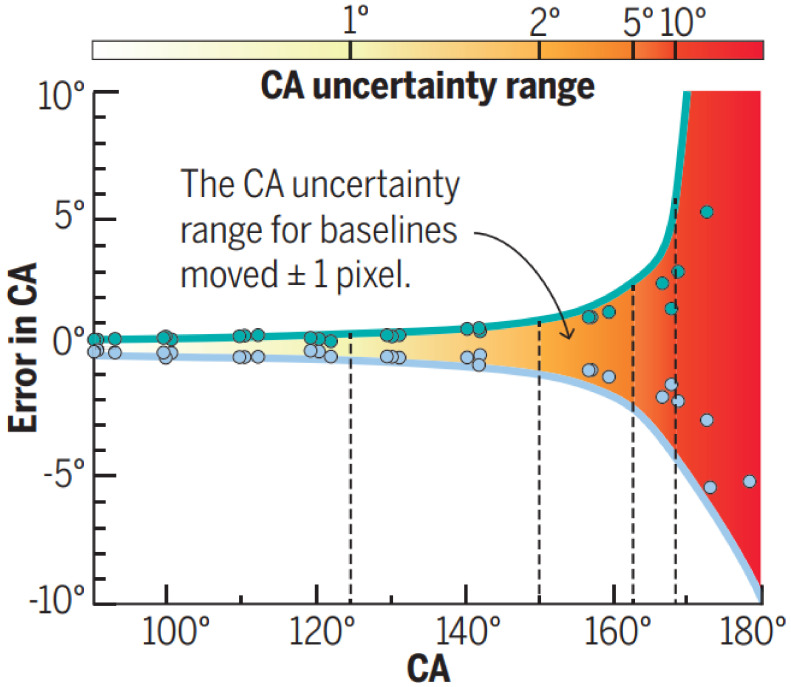
Plot of error in WCA resulting from just 1-pixel error in identifying the correct baseline (i.e., solid/liquid interface). The top curve represents the error for a baseline shifted 1 pixel downward. The bottom curve represents the error for a baseline shifted 1 pixel upward (From [154]. Reprinted with permission from AAAS).

**Figure 25 biomimetics-07-00196-f025:**
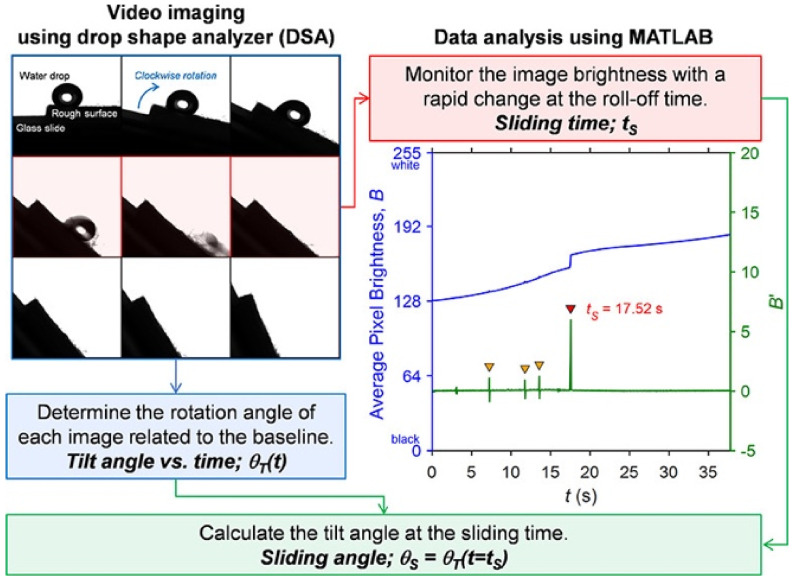
Determination of the SA using an image analyzer aided by a computational algorithm [175].

**Figure 26 biomimetics-07-00196-f026:**
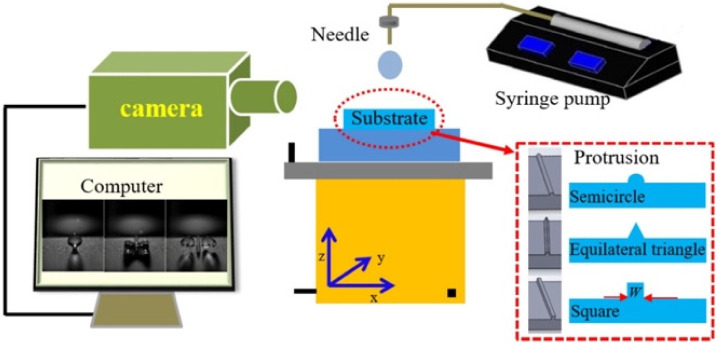
Experimental setup to monitor drop impacts on surfaces with various protrusions (Reprinted from [188] with permission from Elsevier).

**Figure 27 biomimetics-07-00196-f027:**
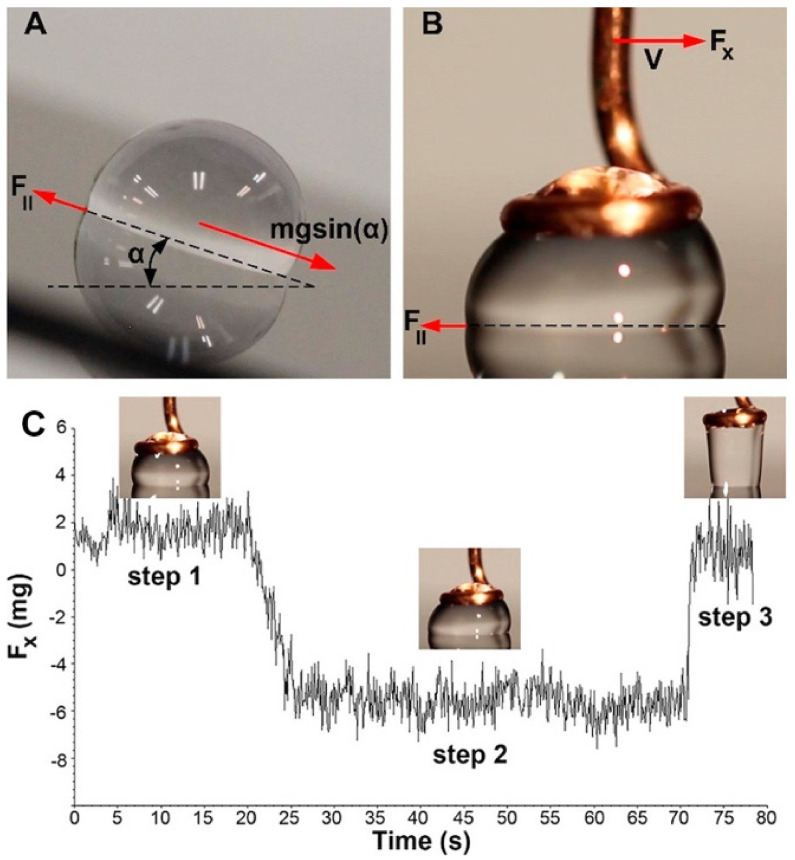
Side-view optical image of 20 µL water drop: (**A**) sliding on a hydrophobic surface at a tilt angle α. When the drop begins to slide, the sliding angle is equal to α; (**B**) sheared on a hydrophobic surface at a velocity V and corresponding applied force F_x_. $F_{∥}$ is the friction force at the interface between the water drop and the surface; and (**C**) plot of friction data between a 20 µL water drop and an OTS-modified hydrophobic silicon surface collected using a nanotribometer at various stages in the measurement; step 1—Approach, step 2—Shear, step 3—Retract and move to start position. Corresponding optical images of the water drops in contact with the surface during the various steps are also shown as insets [174].

**Figure 28 biomimetics-07-00196-f028:**
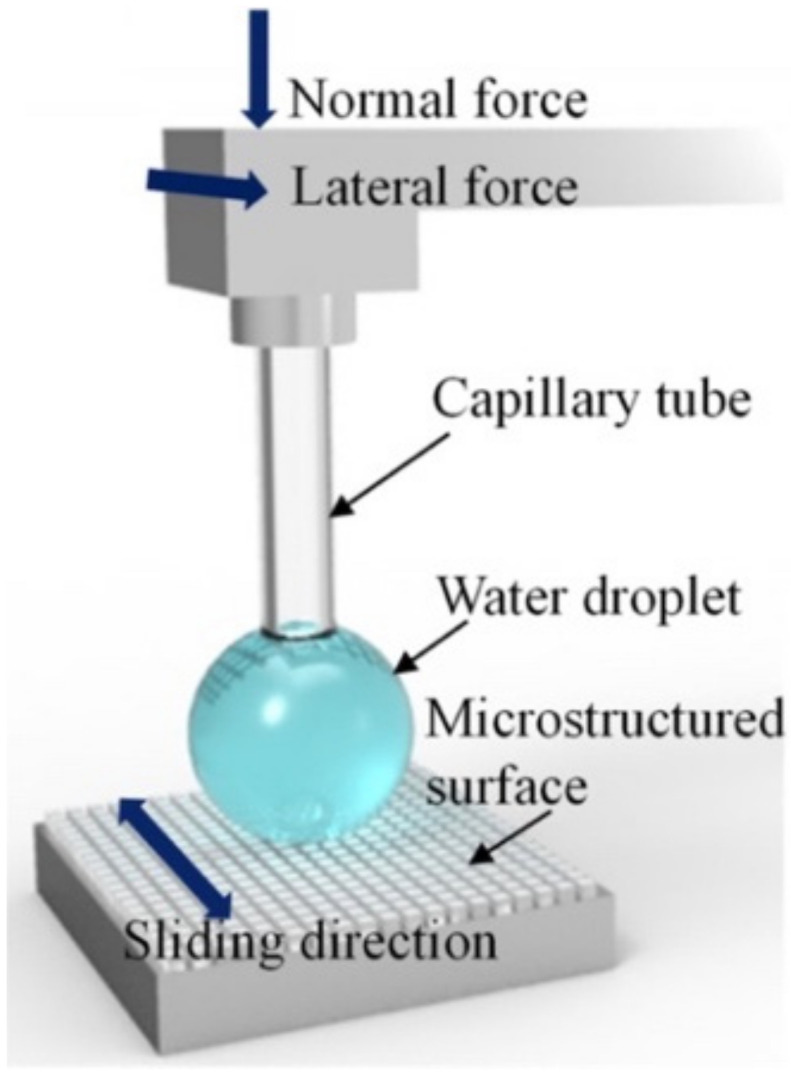
Schematic illustration of the use of a nanotribometer with a capillary tube to measure the sliding friction force between a drop and a surface (Reprinted with permission from [198]. Copyright {2017} American Chemical Society).

**Figure 29 biomimetics-07-00196-f029:**
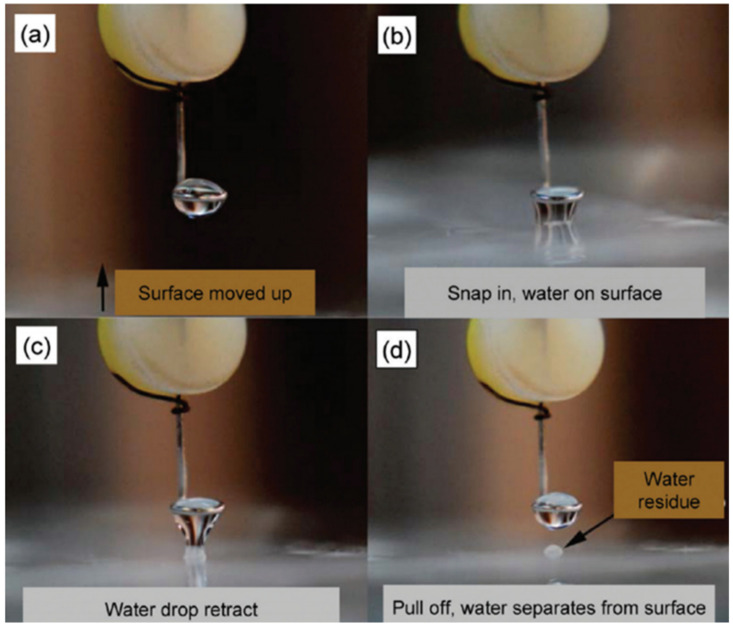
Optical images of the snap-in and pull-off force measurements by the microelectromechanical balance system (Reprinted with permission from [212]. Copyright {2011} American Chemical Society).

**Figure 30 biomimetics-07-00196-f030:**
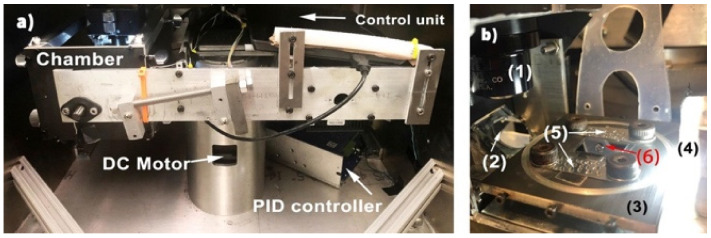
Optical images of the (**a**) centrifugal Adhesion Balance (CAB); and (**b**) individual components inside the instrument including the (1) camera lens, (2) mirrors (45° tilted), (3) sample stage, (4) light source, (5) satellite drops, and (6) central drop (Reprinted with permission from [206]. Copyright {2019} American Chemical Society).

**Figure 31 biomimetics-07-00196-f031:**
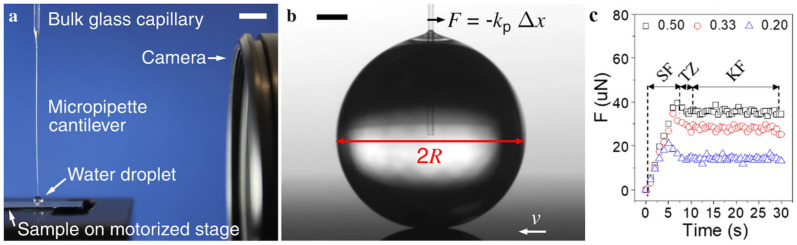
(**a**) Optical image of an MFS setup with a droplet deposited on a SH surface. A water drop is suspended from a force-calibrated micropipette cantilever; (**b**) high magnification image of a water drop attached to a capillary tube. While the stage moves, the drop experiences a friction force thereby deflecting the capillary tube [227]; and (**c**) friction force versus time on three microstriped surfaces with different fractional areas. A capillary force sensor is able to differentiate the different phases of the friction including the static friction (SF), transition zone (TZ), and kinetic friction (KF). (Reprinted from [173] with permission from Elsevier).

**Table 1 biomimetics-07-00196-t001:** Summary of advantages and disadvantages of the various techniques used to create superhydrophobic surfaces.

Technique	Advantages	Disadvantages
Spray, spin, dip coating	Simple, fast, cost-efficient, scalable, damage easily repairable, solvent reusability, applicable to wide range of materials	Not environmentally friendly, poor film stability and uniformity
Chemical etching	Simple, fast, cost-efficient, scalable	Use of toxic solutions, poor film uniformity, applicable to metals and alloys mainly
Sol–gel	Cost-efficient, high film purity and homogeneity, tunability of size and morphology of particles in film, scalable, applicable to wide range of materials	Slow process, susceptible to film delamination
Laser processing	Simple, relatively fast, high film uniformity and precision of structures, environmentally friendly, applicable to wide range of materials	Costly, not scalable, requires optimization of several process parameters
Electrodeposition	Cost-efficient, tunability of texture morphology, scalable	Applicable to metals mainly
Electrospinning	Fiber formation yielding large surface area, film homogeneity	Limited control of porosity, use of organic solvents, relatively slow, limited to polymer fibers
Chemical vapor deposition	Residue-free, film homogeneity, applicable to wide range of materials	Costly, High T and P, not scalable, requires precise control of operating parameters
Lithography	Cost-efficient, controllable shape and size of structures, reusability of templates, environmentally friendly, applicable to wide range of materials	May require cleanroom, not scalable, multi-step process, requires a flat substrate
Thermal deposition	Scalable, high film reproducibility, applicable to wide range of materials	Multi-step process required for large-scale production
Plasma etching	High aspect ratio structures, applicable to wide range of materials	Costly, potential toxic gas formation

**Table 2 biomimetics-07-00196-t002:** List of typical experimental parameters used in drop impact studies.

Reference	Drop Volume (µL)	Contact Angle (°)	Contact Time (ms)	Impact Velocity (m/s)	Weber Number	Camera Speed (fps)
Liu et al. [192]	6 and 13 µL	>165	16.5	0.59–1.72	7.1–58.5	10,000
Liu et al. [193]	13 µL	165 ± 2.9	3.8	-	3.9–23.5	10,000
Chen et al. [194]	4 and 7 µL	107 ± 0.8	-	0.4, 0.48	3.2, 3.8	20,000
Liu et al. [195]	13 µL	163.4 ± 2.6	11.8	0.63	7.9	10,000

**Table 3 biomimetics-07-00196-t003:** List of techniques, reported sensitivities, and typical drop volumes used in the cited references.

Reference	Instrument	Sensitivity	Drop Volume (µL)
Liimatainen et al. [209]	Microforce sensing probe	±0.1 nN	1.5
Wang et al. [149]	AFM	±0.1 nN	1
Qiao et al. [198]	Nanotribometer	±3 nN	2
Sun et al. [213]	Micro-electromechanical balance	±1 µN/m	4
Backholm et al. [226]	Micropipette force sensor	±4 nN	~1
Daniel et al. [229]	Droplet force apparatus	±10 nN	~4.2–6.4
Zhang et al. [214]	Micro-electronic balance	±0.03 µN	3
Samuel et al. [212]	Micro-electromechanical balance	±0.1 µN	~5
Cheng et al. [207]	Micro-electromechanical balance	±0.1 µN	4
Gao et al. [217]	Laser deflection system	±1 µN	1.5–8
Ning et al. [211]	Micro-electromechanical balance	±1 µN	5
Daniel et al. [230,231]	Cantilever force sensor	±0.1 µN	1

## Data Availability

Not applicable.

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
