# Peer review of "Advances in the Fabrication and Characterization of Superhydrophobic Surfaces Inspired by the Lotus Leaf"

_biomimetics, 2022, doi:10.3390/biomimetics7040196_

Round 1

Reviewer 1 Report

This manuscript presents a review on advances in the fabrication and characterization of superhydrophobic surfaces inspired by the lotus leaf. However, there are several questions that need to be addressed:

1. “Nanosecond pulse lasers are more cost and time-efficient compared to femtosecond and picosecond pulse lasers but can cause thermal effects including recast layers, cracks, and self-organized structures [96].” This sentence is ambiguous and needs to be revised. In addition, self-organized structures not only exist in nano-second pulse lasers ablated surface, but also in femtosecond and pico-second pulse lasers. Moreover, self-organized structures are not necessarily bad for hydrophobicity. 

2. There are many combined strategies to fabricate superhydrophobic surfaces, it is suggested to summarize them to a separate subsection. 

3. It is suggested to draw an overview figure. 

4. Self-organized structures induced by laser are also important to regulate surface wettability. There are some articles for reference:

[1] Y. Song, C. Wang, er al, Controllable superhydrophobic aluminum surfaces with tunable adhesion fabricated by femtosecond laser, Optics & Laser Technology, 2017, 102: 25-31.

[2] K. Ding, C. Wang, et al, Single-step femtosecond laser structuring of multifunctional colorful metal surface and its origin, Surf. Interfaces 34 (2022) 102386.

[3] K. Ding, C. Wang, et al, Large-area cactus-like micro-/nanostructures with anti-reflection and superhydrophobicity fabricated by femtosecond laser and thermal treatment, Surf. Interfaces 33 (2022) 102292. 

Based on my remarks, I will accept this manuscript for publication after minor revision.

Reviewer 2 Report

The MS presents interesting information compiling about superhydrophobic surfaces, but some order must be done, along the text to allow the reader to have the main idea of the review. Before its consideration for its publication, some comments should be addressed:

1)    The authors should add the 2022-year information in Fig. 2.

2)    Line 62 presents the complete scientific name of the lotus plant. 

3)    The figures' caption should be kept in simpler way. 

4)  InSection 2, an introductory paragraph is needed to explain why the Youngs equation is important in superhydrophobic surfaces. 

5)    Equation numbers should be mentioned in the text. 

6)    Section 3.1 is not well described as the authors mention some lines below, “Surface coating” is not a synthesis or fabrication technique of the superhydrophobic materials, it is a deposition method. The title must be changed or re-arrange the whole section. 

7)    The synthesis methods must be classified in chemical and physical methods. Also, the authors must clarify in section 3, which will be the “superhydrophobic” materials that they will be discussing and why. It would be easier another classification of the materials, and this will help to the reader to get the purpose of the review. 

8)    Present tables 1, 2, and 3 according to the journal template. 

9)    Section 4 title, there is a typo; revise, please. 

10) Please revise the next reference to complete the information about how the synthesis conditions affect the hydrophobicity properties: 

https://doi.org/10.3390/coatings8040120

Round 2

Reviewer 2 Report

No further comments.